# SELF-SUPERVISED POCKET PRETRAINING VIA PROTEIN FRAGMENT-SURROUNDINGS ALIGNMENT

**Bowen Gao[1]\*, Yinjun Jia[2]\*, Yuanle Mo[3], Yuyan Ni[4], Weiying Ma[1], Zhiming Ma[4], Yanyan Lan[1,5]†**

[1]Institute for AI Industry Research, Tsinghua University
[2]School of Life Sciences, IDG/McGovern Institute for Brain Research, Tsinghua University
[3]School of Information and Software Engineering, UESTC
[4]Academy of Mathematics and Systems Science, Chinese Academy of Sciences
[5]Beijing Frontier Research Center for Biological Structure, Tsinghua University

## ABSTRACT

Pocket representations play a vital role in various biomedical applications, such as druggability estimation, ligand affinity prediction, and *de novo* drug design. While existing geometric features and pretrained representations have demonstrated promising results, they usually treat pockets independent of ligands, neglecting the fundamental interactions between them. However, the limited pocket-ligand complex structures available in the PDB database (less than 100 thousand non-redundant pairs) hampers large-scale pretraining endeavors for interaction modeling. To address this constraint, we propose a novel pocket pretraining approach that leverages knowledge from high-resolution atomic protein structures, assisted by highly effective pretrained small molecule representations. By segmenting protein structures into drug-like fragments and their corresponding pockets, we obtain a reasonable simulation of ligand-receptor interactions, resulting in the generation of over 5 million complexes. Subsequently, the pocket encoder is trained in a contrastive manner to align with the representation of pseudo-ligand furnished by some pretrained small molecule encoders. Our method, named ProFSA, achieves state-of-the-art performance across various tasks, including pocket druggability prediction, pocket matching, and ligand binding affinity prediction. Moreover, our work opens up a new avenue for mitigating the scarcity of protein-ligand complex data through the utilization of high-quality and diverse protein structure databases. The code and data is available at https://github.com/bowen-gao/ProFSA.

## 1 INTRODUCTION

In drug discovery, AI-driven methods have made significant strides, with a growing focus on the task of representing protein pockets. These pockets play a pivotal role in binding small molecule ligands through diverse interactions like hydrophobic forces, hydrogen bonds, $\pi$-stacking, and salt bridges (de Freitas & Schapira, 2017). While traditional handcrafted features like curvatures, solvent-accessible surface area, hydropathy, and electrostatics (Guilloux et al., 2009; Vascon et al., 2020) have provided valuable domain insights, their computational demands and inability to capture intricate interactions limit their utility.

In contrast, data-driven deep learning methods, such as the two-tower architectures proposed by Gao et al. (2022), Karimi et al. (2019) and Öztürk et al. (2018), show promise but are constrained by the scarcity of complex structure data (less than 100 thousand non-redundant pairs in BioLip2 database (Zhang et al., 2023)). To address this limitation, self-supervised learning techniques, like the one introduced by Zhou et al. (2023), focus on restoring corrupted data, allowing them to leverage extensive unbounded pocket data for representation learning. However, these methods often overlook the vital ligand-pocket interactions due to the lack of proper supervision signals from the ligand

---

*Equal contirbution
†Correspondence to `lanyanyan@air.tsinghua.edu.cn`

side. To fully exploit the potential of pocket representation learning and emphasize interactions between pockets and ligands, there is a pressing need for the development of large-scale datasets and innovative pretraining methods.

To address the aforementioned challenges, we introduce an innovative methodology that utilizes a protein fragment-surroundings alignment technique for large-scale data construction and pocket representation learning. The fundamental principle of this approach is the recognition that protein-ligand interactions adhere to the same underlying physical principles as inter-protein interactions, due to their shared organic functional groups, like phenyl groups in the phenylalanine and carboxyl groups in the glutamate. Therefore, we could curate a larger and more diverse dataset by extracting drug-like fragments from protein chains to simulate the protein-ligand interaction and learn ligand-aware pocket representations with the guidance of pretrained small molecule encoders, which have been extensively explored by many previous works by using large scale small molecule datasets, e.g. Uni-Mol (Zhou et al., 2023), Frad (Feng et al., 2023).

To bridge the gap between fragment-pocket pairs and ligand-pocket pairs, we employ three strategies. Firstly, we focus on residues with long-range interactions, excluding those constrained with peptide bonds absent in real ligand-protein complexes (Wang et al., 2022a). Secondly, we sample fragment-pocket pairs based on their relative buried surface area (rBSA) and the joint distribution of pocket and ligand sizes to mimic real ligands. Thirdly, we address property mismatches caused by the segmentation process by applying terminal corrections (Marino et al., 2015; Arbour et al., 2020) to fragments. This results in a dataset of 5.5 million pairs from the PDB database.

With this dataset, we introduce a molecular-guided contrastive learning method to obtain ligand-aware pocket representations. Our approach trains the pocket encoder to differentiate between positive ligands and other negative samples in the same batch. Our contrastive framework is capable of encoding diverse interaction patterns into pocket representations regardless of inconsistency between protein fragments and real ligands. To counteract biases from peptide fragments used as ligands, we align the pocket encoder with a pretrained small molecule encoder, whose weights are fixed. This pretrained encoder serves as a guide, transferring binding-specific and biologically relevant information from extensive molecular datasets to our pocket encoder.

Our contributions are articulated as follows:

1. The proposal of a novel scalable pairwise data synthesis pipeline by extracting pseudo-ligand-pocket pairs from protein-only data, which has the potential to be applied to much larger predicted structure data generated by AlphaFold (Jumper et al., 2021) or ESMFold (Lin et al., 2023).

2. The introduction of a new molecular guided fragment-surroundings contrastive learning method for pocket representations, which naturally distillates comprehensive structural and chemical knowledge from pretrained small molecule encoders.

3. The achievement of significant performance boosts in various downstream applications, including tasks that solely require pocket data, as well as those involving pocket-ligand pair data, underscoring the potential of ProFSA as a powerful tool in the drug discovery field.

## 2 RELATED WORK

### 2.1 POCKET PRETRAINING DATA

Currently available protein pocket data are all collected from the Protein Data Bank (PDB) (Berman et al., 2000). The most famous database is the PDBBind (Liu et al., 2015; Wang et al., 2005; 2004), which consists of 19,443 protein-ligand pairs in the latest version (v2020). While the PDBBind database focuses on protein-ligand pairs with available affinity data, other databases also collect pairs without affinity data. Biolip2 (Zhang et al., 2023) is one of the most comprehensive ones, which includes 467,808 pocket-ligand pairs, but only 71,178 pairs are non-redundant (the weekly version of 2023-09-21). Due to the expensive and time-consuming nature of structural biology experiments, it is not feasible to expect significant growth in these types of databases in the near future. On the other hand, individuals may rely on pocket prediction softwares, such as Fpocket (Zhou et al., 2023), to expand the pocket dataset. However, pockets identified by these programs inherently lack paired ligands, thus resulting in insufficient knowledge of protein-ligand interactions.

## 2.2 POCKET PRETRAINING METHODS

Recently, several pretraining methods have demonstrated exceptional performance in creating effective representations of protein pockets that can be utilized across a range of downstream tasks. Some methods view the binding site as a part of protein targets and directly pretrain on protein structures(Liu et al., 2023; Wu et al., 2022) or sequences(Karimi et al., 2019). Other methods operate on explicit pockets isolated from a target to make the model more focused. One such method is Uni-Mol (Zhou et al., 2023), which offers a universal 3D molecular pretraining framework and involves a pocket model that has been pretrained on 3 million pocket data. However, this method lacks a specific focus on the interactions between pockets and ligands. Additionally, CoSP (Gao et al., 2022) employs a co-supervised pretraining framework to learn both pocket and ligand representations simultaneously. Nevertheless, the learned representations are not entirely satisfactory due to the lack of diversity in the chemical space caused by the insufficient data.

## 2.3 MOLECULE PRETRAINING METHODS

Molecular representation learning greatly facilitates drug discovery in tasks like molecular property prediction and drug-target interaction prediction. Following the huge success of self-supervised learning in natural language processing (Devlin et al., 2019) and computer vision (He et al., 2021; Chen et al., 2020), various pre-training methods have been developed to address the scarcity of labeled data in this field. Initially, most methods operate on 1D SMILES strings (Wang et al., 2019) or 2D graphs (Hu et al., 2020) with masked language models, self-contrastive objectives, or alignments between 1D and 2D representations (Lin et al., 2022). However, recognizing the importance of a molecule's 3D geometric structure in determining its physical and chemical properties, recent approaches increasingly focus on using 3D molecular data for pre-training (Zhou et al., 2023; Feng et al., 2023; Zaidi et al., 2023), offering a more comprehensive understanding of molecular structures.

# 3 OUR APPROACH

## 3.1 CONSTRUCTING PSEUDO-LIGAND-POCKET COMPLEXES FROM PROTEIN DATA

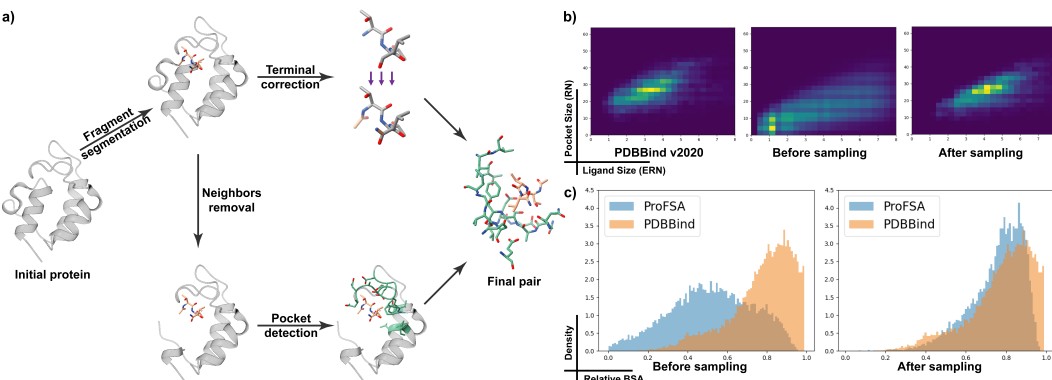

Figure 1: **a)** The pipeline for isolating pocket-ligand pairs from proteins; **b)** Joint distributions of the pocket size and the ligand size of the PDBBind dataset, our ProFSA dataset before stratified sampling and after stratified sampling, respectively; **c)** Comparations between the ProFSA dataset and the PDBBind dataset in terms of distributions of rBSA of ligand-pocket pairs.

To address the scarcity of experimentally determined pocket-ligand pairs, we've developed an innovative strategy for mining extensive protein-only data in the PDB repository. We reason that knowledge about non-covalent interactions could be generalized from proteins to small molecules. Therefore, we extract fragments from a protein structure that closely resembles a ligand and designate the surroundings as the associated pocket of this pseudo-ligand.

Our data construction process starts with non-redundant PDB entries clustered at a 70 percent sequence identity threshold. This process comprises two phases: pseudo-ligand construction and

pocket construction, as illustrated in Figure 1. In the first phase, we iteratively isolate fragments ranging from one to eight residues, from the N terminal to the C terminal, avoiding any discontinuous sites or non-standard aminoacids. To address biases introduced by breaking peptide bonds during fragment segmentation, we apply terminal corrections, specifically acetylation and amidation (Marino et al., 2015; Arbour et al., 2020), to the N-terminal and C-terminal, respectively, resulting in the final pseudo-ligands. In the second phase, we exclude the five nearest residues on each side of the acquired fragment, focusing on long-range interactions. We then designate the surrounding residues containing at least one heavy atom within a 6Å proximity (following the setup of Uni-Mol, other cutoff values are presented in Table 10) to the fragment as the pocket. This process yields pairs of finalized pockets and pseudo-ligands.

Ligand-pocket pairs generated with the aforementioned procedure can mimic most non-covalent interactions. As we visualize in Figure 2, hydrogen bonds are formed within protein backbones and polar amino acids like serine or histidine; $\pi-\pi$ stackings are common between aromatic amino acids including tyrosine, tryptophan, and phenylalanine; also salt bridges can be expected between oppositely charged amino acids like arginine and glutamate. Some other rare interaction types are not included in the figure, but they also exist in the generated dataset. For example, anion-$\pi$ interactions can exist between positively charged arginine and phenylalanine; even some non-canonical interactions like C-H$\cdots\pi$ have been reported (Brandl et al., 2001). Therefore, as observed by Polizzi & DeGrado (2020), intra-protein interactions can be a good simulation for protein-ligand interactions.

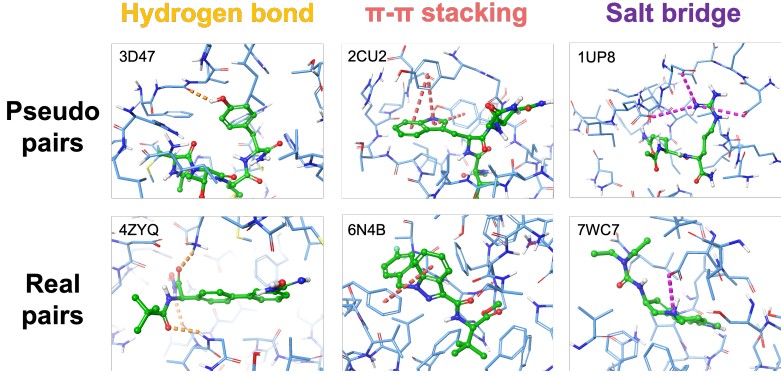

Figure 2: Visualization of common interaction types presented in both pseudo pairs and real pairs.

Pseudo-complexes are sampled to approximate distributions of the PDBBind dataset. The resulting dataset has a similar distribution as the PDBBind, in terms of rBSA and the joint distribution of pocket-ligand size, as shown in Figure 1b and c. More details are provided in Appendix A.

### 3.2 CONTRASTIVE LEARNING IN POCKET-FRAGMENT SPACE

We propose a contrastive learning approach in the protein-fragment space to infuse pocket representations with interaction knowledge. As previously demonstrated (Wang & Isola, 2020), the capacity to attain feature alignment from positive data is a vital characteristic of contrastive learning. Previous research indicates that contrastive learning is effective in both image-text (Radford et al., 2021) and pocket-ligand interaction (Gao et al., 2024).

Specifically, for a protein pocket $p$ and its corresponding pseudo-ligand $l$, we denote their normalized encoding vectors as $s$ and $t$. Let $g_T$ and $g_S$ denote the ligand and protein prediction networks, respectively. Then two distinct contrastive losses are used to facilitate the training process:

$$\mathcal{L}_1(\boldsymbol{t}, \boldsymbol{s}) = -\log \frac{e^{g_T(\boldsymbol{t}) \cdot g_S(\boldsymbol{s})}}{\sum_{\boldsymbol{s}_i \in \mathbb{N}_s \cup \{\boldsymbol{s}\}} e^{g_T(\boldsymbol{t}) \cdot g_S(\boldsymbol{s}_i)}} = -g_T(\boldsymbol{t}) \cdot g_S(\boldsymbol{s}) + \log \sum_{\boldsymbol{s}_i \in \mathbb{N}_s \cup \{\boldsymbol{s}\}} e^{g_T(\boldsymbol{t}) \cdot g_S(\boldsymbol{s}_i)}, \quad (1)$$

$$\mathcal{L}_2(\boldsymbol{t}, \boldsymbol{s}) = -\log \frac{e^{g_T(\boldsymbol{t}) \cdot g_S(\boldsymbol{s})}}{\sum_{\boldsymbol{t}_i \in \mathbb{N}_t \cup \{\boldsymbol{t}\}} e^{g_T(\boldsymbol{t}_i) \cdot g_S(\boldsymbol{s})}} = -g_T(\boldsymbol{t}) \cdot g_S(\boldsymbol{s}) + \log \sum_{\boldsymbol{t}_i \in \mathbb{N}_t \cup \{\boldsymbol{t}\}} e^{g_T(\boldsymbol{t}_i) \cdot g_S(\boldsymbol{s})}, \quad (2)$$

where $\mathbb{N}_s$ and $\mathbb{N}_t$ refer to the in-batch negative samples for protein pockets and pseudo-ligands, respectively. The primary purpose of the first loss is to identify the true protein pocket from a batch of

samples when given a pseudo-ligand. Similarly, the second loss seeks to identify the corresponding ligand fragment for a given pocket.

Hence, the ultimate loss utilized in our training process is:

$$L_{CL} = E_{p(\boldsymbol{t}, \boldsymbol{s})}[\mathcal{L}_1(\boldsymbol{t}, \boldsymbol{s}) + \mathcal{L}_2(\boldsymbol{t}, \boldsymbol{s})]. \tag{3}$$

Our method stands out from traditional contrastive learning approaches by keeping the molecule encoder's parameters constant (Figure 3, the left panel). Although the dataset we generated is carefully aligned with the PDBBind dataset, it is worth noticing that pseudo ligands are still different from real ligands in terms of chemical properties like logP and rotatory bond numbers (Figure 9). Fixed molecule encoders could ease the discrepancy between real ligands and pseudo ligands, allow us to distill learned biochemistry knowledge from large pretrained molecule encoders, and further reduce computational costs. Thus pocket representations aligned with the fixed fragment representations also encode biologically meaningful properties (Figure 3, the right panel). Empirical proof of the advantages of fixed encoders is shown in Table 5, and the corresponding theorem and the rigorous proof are shown in Appendix B.

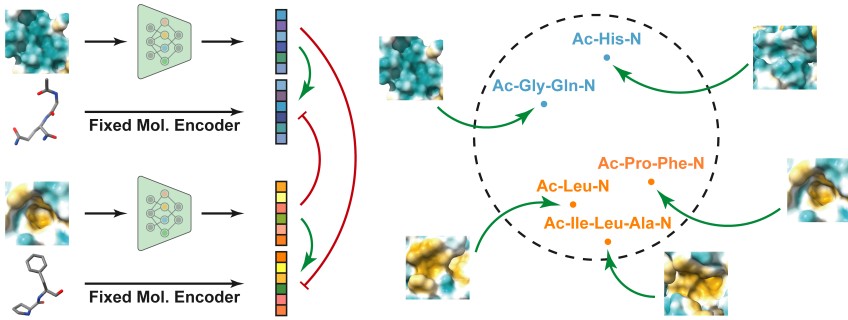

Figure 3: An illustration of protein fragment-surroundings alignment framework. Pockets are encoded by our pocket encoder, which is trained to align with fragment representations given by fixed pretrained molecule encoders. A simplified hydropathy-related (indicated by blue or orange color) example illustrates that fragment properties recognized by pretrained molecule encoders could guide pocket representation learning.

# 4 EXPERIMENTS

To evaluate the performance of our proposed model, we conduct extensive experiments mainly from three perspectives: **1. Pocket-only tasks** including pocket druggability prediction and pocket matching in § 4.1 and § 4.2; **2. The pocket-molecule task** of ligand binding affinity prediction in § 4.3; **3. Ablation studies** in § 4.4 illustrating the impact of diverse data scales, molecular encoders, and data distributions. Detailed experiment settings are provided in Appendix E. The default pocket and molecular encoders used in ProFSA are based on Uni-Mol architectures. The molecule encoder and its associated weight are directly obtained from the official Uni-Mol repository, whereas the pocket encoder is trained from scratch. Detailed information on encoder architectures is provided in Appendix D.

## 4.1 POCKET DRUGGABILITY PREDICTION

**Experimental Configuration** We evaluate ProFSA's ability to predict the physical and pharmaceutical properties of pockets on the druggability prediction dataset proposed by Uni-Mol. This dataset consists of 4 different tasks, namely the Fpocket score, Druggability score, Total SASA, and Hydrophobicity score. Since all of them are regression tasks, we use the Root Mean Square Error (RMSE) as the evaluation metric.

**Baselines** We have included both the finetuning results and the zero-shot results for ProFSA. Following Wu et al. (2018), K-nearest neighbors regression is utilized to produce the zero-shot prediction results.

**Results** As illustrated in Table 1, ProFSA outclasses Uni-Mol in both the finetuning and zero-shot settings. Notably, the margin of superiority is more pronounced in the zero-shot scenario, underscoring our model's efficacy in handling out-of-distribution data. This suggests that during the pretraining phase, rich druggability related information is effectively transferred from the molecular encoder to our specialized pocket encoder.

Table 1: Druggability prediction results using the RMSE metric.

|  |  | Fpocket ↓ | Druggability ↓ | Total Sasa ↓ | Hydrophobicity ↓ |
|---|---|---|---|---|---|
| Finetuning | Uni-Mol | 0.1140 | 0.1001 | 20.73 | 1.285 |
|  | ProFSA | **0.1077** | **0.0934** | **20.01** | **1.275** |
| zero-shot | Uni-Mol | 0.1419 | 0.1246 | 49.00 | 17.03 |
|  | ProFSA | **0.1228** | **0.1106** | **30.50** | **13.07** |

## 4.2 POCKET MATCHING

In addition to druggability prediction, we conduct experiments on the protein pocket matching task, also known as binding site comparison, which plays a fundamental role in drug discovery.

**Experimental Configuration** Two datasets are commonly utilized for this task: the Kahraman dataset (Kahraman et al., 2010; Ehrt et al., 2018), which helps determine whether two non-homologous proteins bind with the same ligand, and the TOUGH-M1 dataset (Govindaraj & Brylinski, 2018), which involves the relaxation of identical ligands to identify similar ones. The Kahraman dataset consists of 100 proteins binding with 9 different ligands, and we adopt a reduced dataset removing 20 $PO_4$ binding pockets due to the low number of interactions (Ehrt et al., 2018), while the TOUGH-M1 dataset is comprised of 505,116 positive and 556,810 negative protein pocket pairs that are defined from 7524 protein structures.

All the data splitting, training and evaluation strategies used in this paper follow the practice of the previously published work DeeplyTough (Simonovsky & Meyers, 2020). The computation of pocket similarity in ProFSA involves utilizing the cosine similarity between two pocket representations. We report AUC-ROC results for both the finetuning and the zero-shot settings.

**Baselines** Our baseline models consist of diverse binding site modeling approaches, including the shape-based method PocketMatch (Yeturu & Chandra, 2008), the grid-based methods such as DeeplyTough (Simonovsky & Meyers, 2020), the graph-based method IsoMIF (Chartier & Najmanovich, 2015), existing softwares such as SiteEngine (Shulman-Peleg et al., 2005) and TM-align (Zhang & Skolnick, 2005). We also include pretraining approaches, such as Uni-Mol and CoSP.

Table 2: Pocket matching results using the AUC metric.

|  | Methods | Kahraman(w/o $PO_4$) ↑ | TOUGH-M1 ↑ |
|---|---|---|---|
| Traditional | TM-align | 0.66 | 0.65 |
|  | SiteEngine | 0.64 | 0.73 |
|  | PocketMatch | 0.66 | 0.64 |
|  | IsoMIF | 0.75 | - |
| Zero-shot | Uni-Mol | 0.66 | 0.76 |
|  | ProFSA | **0.80** | **0.82** |
| Finetuning | DeeplyTough | 0.67 | 0.91 |
|  | CoSP | 0.62 | - |
|  | Uni-Mol | 0.71 | 0.79 |
|  | ProFSA | **0.85** | **0.94** |

**Results** We utilized results from Simonovsky & Meyers (2020) and Gao et al. (2022) for most baseline methods, while implementing Uni-Mol ourselves. As indicated in Table 2, our model significantly outperforms others in both finetuning and zero-shot settings, surpassing learning methods like Uni-Mol and CoSP, as well as traditional methods like SiteEngine and IsoMIF, across Kahraman and TOUGH-M1 datasets. Importantly, while other finetuned methods couldn't exceed IsoMIF's performance on the Kahraman dataset, which assesses pocket similarity based on accommodating

diverse ligands, our zero-shot learning model notably did. This underscores our model's strong ability to internalize relevant interaction features, even though it's only pretrained on protein structural data.

In the Kahraman task, our model outperformed Uni-Mol within the estradiol binding pocket class (AUROC scores: ProFSA vs. UniMol - 0.947 vs. 0.647). To illustrate the impact of interaction-aware pretraining on pocket representation, we visualize a case where Uni-Mol struggled to recognize the similarity. Sex hormone binding globulin (PDB:1LHU) and estrogen receptor (PDB:1QKT) exhibit zero sequence similarity but all bind to estradiol. Querying with 1LHU, our model recovered 1QKT with a z-score of 2.22, while Uni-Mol yielded 0.22. Despite distinct folding architectures and limited geometric similarity in their binding pockets, both receptors shared a parallel interaction pattern (Figure 4a). Our contrastive training prioritizes key residues for inter-molecule interactions, while Uni-Mol relies on local geometry (e.g., secondary structures) to fulfill the denoising objective. Therefore, such models could be potentially blinded to fine interaction patterns.

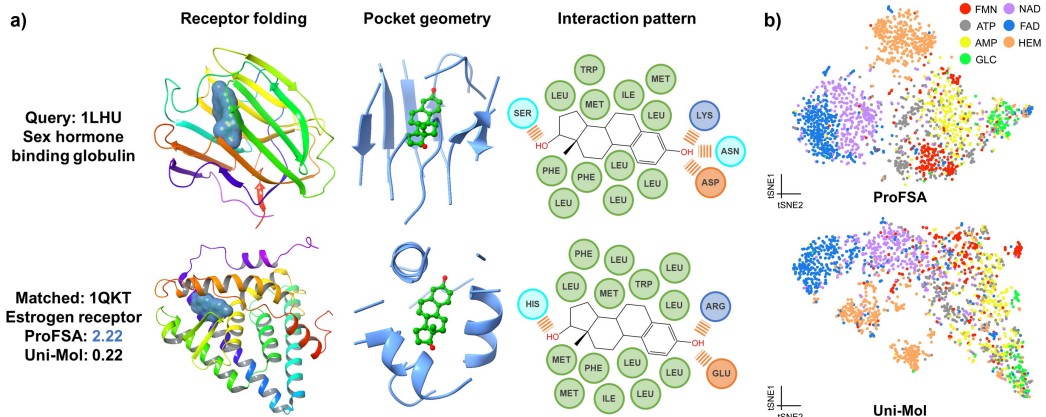

Figure 4: **a)** Visualization of two estradiol binding proteins that ProFSA performs better on. Positively charged, negatively charged, polar, and hydrophobic amino acids are represented in different colors to visualize interaction patterns. Hydrogen bonds are represented by dashed lines. **b)** A t-SNE visualization of pretrained representations of 7 types of ligand binding pockets collected from the BioLip database. Our ProFSA model distinguished FMN, ATP, AMP, and GLC binding pockets better compared with the Uni-Mol model.

To better showcase the acquired pocket representations, we perform visualization based on the collected pockets from the BioLip database (Zhang et al., 2023). Following a similar procedure as with the Kahraman dataset, we select seven frequently encountered ligands, namely NAD, FAD, HEM, FMN, ATP, and GLC. As depicted in Figure 4b, in which both ProFSA and Uni-Mol demonstrate good separation for NAD, FAD, and HEM (indicated in purple, blue, and orange respectively), ProFSA outperforms the Uni-Mol encoder in distinguishing pockets that bind FMN, ATP, AMP, and GLC (highlighted in red, grey, yellow, and green respectively). This finding is remarkably significant as the differences among these four ligands are exceedingly subtle. For instance, when comparing ATP and AMP, only two phosphate groups are absent in AMP; GLC shares similarities with the ribose portion of both ATP and AMP; and FMN exhibits a similar binding pattern with ATP and AMP in terms of $\pi$-$\pi$ stacking (Meyer et al., 2003). In light of these similarities, our results validate that pretraining with fragment-surroundings alignment can assist in learning more intricate interaction preferences compared to previous mask-predict and denoising paradigms.

### 4.3 LIGAND BINDING AFFINITY PREDICTION

Although pockets can exist independently of specific ligands, the concept of pocket is strongly associated with drug-like molecules within the field of drug discovery. As a result, testing the learned pocket representation with regard to binding affinity prediction holds significant meaning.

**Experimental Configuration** We are utilizing the well-acknowledged PDBBind dataset(v2019) for the ligand binding affinity (LBA) prediction task, and we follow strict 30% or 60% protein sequence-identity data split and preprocessing procedures from the Atom3D (Townshend et al.,

2022). Employing such a strict split is crucial as it provides more reliable and meaningful comparisons in terms of the robustness and generalization capability of models.

Given that binding affinity prediction is inherently a regression task, we have integrated a Multi-Layer Perceptron (MLP) module as the regression head for our ProFSA or Uni-Mol. The MLP head operates by taking as input the concatenated forms of both the pocket representation and the molecule representation, and **pretrained encoders are kept fixed to ensure the finetuning outcome closely reflects the quality of the pretraining**.

Table 3: Results on LBA prediction task. Results ranking first and second are highlighted in **bold** and underlined respectively. For detailed standard deviation, kindly refer to Appendix C.1.

| Method | | Sequence Identity 30% | | | Sequence Identity 60% | | |
|---|---|---|---|---|---|---|---|
| | | RMSE ↓ | Pearson ↑ | Spearman ↑ | RMSE ↓ | Pearson ↑ | Spearman ↑ |
| Sequence Based | DeepDTA | 1.866 | 0.472 | 0.471 | 1.762 | 0.666 | 0.663 |
| | B&B | 1.985 | 0.165 | 0.152 | 1.891 | 0.249 | 0.275 |
| | TAPE | 1.890 | 0.338 | 0.286 | 1.633 | 0.568 | 0.571 |
| | ProtTrans | 1.544 | 0.438 | 0.434 | 1.641 | 0.595 | 0.588 |
| Structure Based | Holoprot | 1.464 | 0.509 | 0.500 | 1.365 | 0.749 | 0.742 |
| | IEConv | 1.554 | 0.414 | 0.428 | 1.473 | 0.667 | 0.675 |
| | MaSIF | 1.484 | 0.467 | 0.455 | 1.426 | 0.709 | 0.701 |
| | ATOM3D-3DCNN | 1.416 | 0.550 | 0.553 | 1.621 | 0.608 | 0.615 |
| | ATOM3D-ENN | 1.568 | 0.389 | 0.408 | 1.620 | 0.623 | 0.633 |
| | ATOM3D-GNN | 1.601 | 0.545 | 0.533 | 1.408 | 0.743 | 0.743 |
| | ProNet | 1.463 | 0.551 | 0.551 | 1.343 | **0.765** | 0.761 |
| Pretraining Based | GeoSSL | 1.451 | 0.577 | 0.572 | - | - | - |
| | DeepAffinity | 1.893 | 0.415 | 0.426 | - | - | - |
| | EGNN-PLM | 1.403 | 0.565 | 0.544 | 1.559 | 0.644 | 0.646 |
| | Uni-Mol | 1.520 | 0.558 | 0.540 | 1.619 | 0.645 | 0.653 |
| | ProFSA | **1.377** | **0.628** | **0.620** | **1.334** | 0.764 | **0.762** |

**Baselines** We select a broad range of models as the baselines, including sequence-based methods such as DeepDTA (Öztürk et al., 2018), B&B (Bepler & Berger, 2019), TAPE(Rao et al., 2019) and ProtTrans (Elnaggar et al., 2021); structure-based methods such as Holoprot (Somnath et al., 2021), IEConv (Hermosilla et al., 2021), MaSIF (Gainza et al., 2020), ATOM3D-3DCNN, ATOM3D-ENN, ATOM3D-GNN (Townshend et al., 2022) and ProNet (Wang et al., 2022b); as well as pretraining methods such as GeoSSL (Liu et al., 2023), EGNN-PLM (Wu et al., 2022), DeepAffinity (Karimi et al., 2019) and Uni-Mol (Zhou et al., 2023).

**Results** Most of the baseline results are taken from Wang et al. (2022b), Townshend et al. (2022) and Liu et al. (2023), while the ones of Uni-Mol are implemented on our own. Based on the empirical results presented in Table 3, ProFSA demonstrates a marked advantage over existing approaches, especially at a 30% sequence identity threshold. Across a range of evaluation metrics—RMSE, Pearson correlation, and Spearman correlation—our method consistently outperforms all baselines with or without pretraining. At a 60% threshold, ProFSA still leads in RMSE and Spearman metrics, while achieving a second-place position in Pearson correlation.

It's intriguing to note that methods like ProNet and Holoprot can match our results at the 60% sequence identity but lag considerably at the stricter 30% threshold. This suggests that ProFSA exhibits better robustness and generalization capabilities, especially when there's a pronounced disparity between the training and testing data distributions.

## 4.4 ABLATION STUDY AND ANALYSIS

**Different molecule encoders** ProFSA employs a pretrained molecular encoder to facilitate knowledge transfer from molecular structures to pocket representations. To evaluate its versatility, we tested ProFSA with two alternate encoders: PTVD (Pretraining via Denoising)(Zaidi et al., 2023), which parallels Uni-Mol in its denoising approach, and Frad (Fractional Denoising Method) (Feng et al., 2023), focusing on denoising coordinate noise to learn the anisotropic force field. Utilizing a dataset of 1.1 million for pretraining, ProFSA consistently surpassed the baseline in various benchmarks with different molecular encoders, as illustrated in Table 4. This demonstrates ProFSA's effectiveness and adaptability beyond the Uni-Mol encoder.

Table 4: Comparison analysis of employing different molecular encoders.

| | Druggability Score (Zero-Shot) ↓ | | | | Pocket Matching ↑ | |
|---|---|---|---|---|---|---|
| | Fpocket | Druggability | Total Sasa | Hydrophobicity | Kahraman(w/o PO$_4$) | Tough M1 |
| ProFSA-Uni-Mol | 0.1238 | 0.1090 | 31.17 | 12.01 | 0.7870 | 0.8178 |
| ProFSA-FRAD | 0.1221 | 0.1058 | 31.70 | 11.08 | 0.7401 | 0.7703 |
| ProFSA-PTVD | 0.1222 | 0.1057 | 32.70 | 11.42 | 0.7583 | 0.7548 |
| Uni-Mol | 0.1419 | 0.1246 | 49.00 | 17.13 | 0.6636 | 0.7557 |

**The significance of distributional alignment**    In our data creation process, we use distribution alignment to ensure pseudo-ligands closely match the PDBbind dataset ligands. To show the alignment's impact, we compare the performance of encoders trained on 1.1 million unaligned and aligned datasets. The results, shown in Table 5, highlight a significant performance difference, underscoring the importance of data alignment for optimal model performance.

Table 5: Impact of distributional alignment and fixed molecular encoder.

| | Druggability Score (Zero-Shot) ↓ | | | | Pocket Matching ↑ | |
|---|---|---|---|---|---|---|
| | Fpocket | Druggability | Total Sasa | Hydrophobicity | Kahraman(w/o PO$_4$) | Tough M1 |
| ProFSA | 0.1238 | 0.1090 | 31.17 | 12.01 | 0.7870 | 0.8178 |
| w/o Alignment | 0.1265 | 0.1108 | 34.79 | 14.86 | 0.7614 | 0.7589 |
| w/o fixed mol encoder | 0.1247 | 0.1094 | 32.17 | 12.20 | 0.6905 | 0.7337 |

**Scales of pretraining data**    To investigate how the model's performance is affected by the dataset size, we conduct a series of experiments using pretraining data of different sizes. Our initial pretraining dataset consists of roughly 5.5 million data points. Additionally, we run experiments with reduced dataset sizes, ranging from 44 thousand to 5.5 million data points, to test the scalability and effectiveness of our approach. Our results on both pocket matching and druggability prediction, as demonstrated in Figure 5, exhibit a positive correlation between training data size and the quality of results in most cases, which further highlights the scalability potential of our methodology.

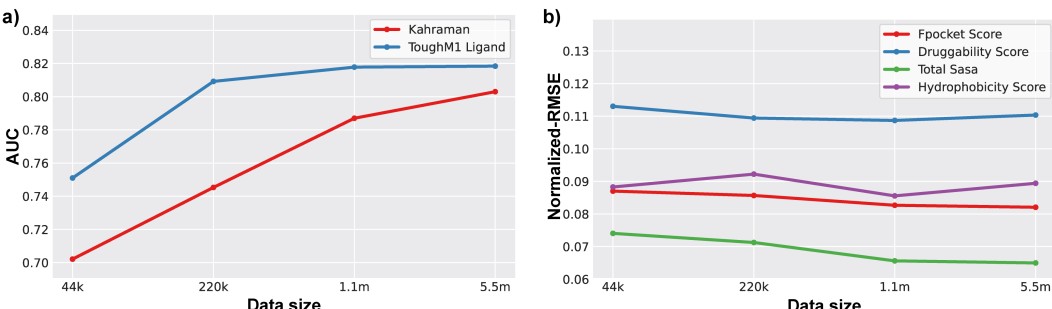

Figure 5: Comparation of different pretraining data sizes. **a)** Results of pocket matching; **b)** Results of druggability prediction.

## 5    CONCLUSION

In this paper, we propose a pioneering approach to tackle the challenges of data scalability and interaction modeling in pocket representation learning, through protein fragment-surroundings alignment. Our methodology involves constructing over 5 million complexes by isolating protein fragments and their corresponding pockets, followed by introducing a contrastive learning method to align the pocket encoder with a pretrained molecular encoder. Our model achieves unprecedented performance in experimental finetuning and zero-shot results across various tasks, including pocket druggability prediction, pocket matching, and ligand binding affinity prediction. We posit that beyond pocket pretraining, our approach has the potential to be adapted for larger predicted structure data, such as data generated by AlphaFold2 and ESMFold, and other structural biology tasks, such as modeling protein-protein interactions, thereby contributing significantly to the advancements in the field.

ACKNOWLEDGEMENT

This work is supported by the National Key R&D Program of China No.2021YFF1201600 and Beijing Frontier Research Center for Biological Structure.

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

## A DETAILS OF DATA CREATION

The dataset is curated using non-redundant PDB entries, which are clustered at a sequence identity threshold of 70%. More specifically, the protein sequences of chains from all PDB entries up to April 28th, 2023, are sourced from the RCSB PDB mirror website. Employing the easycluster

tool in the MMseqs2 software, these protein sequences undergo clustering at a 70% sequence identity threshold. This process culminates in the identification of a total of 53,824 unique PDB IDs from cluster centers. Furthermore, only the primary bioassembly corresponding to each PDB ID is retrieved for the following procedures.

For each PDB file, each residue is carefully examined to eliminate non-standard amino acids, non-protein residues, or other corner cases from the structure. In cases where alternative locations are present, only the first instance is retained. Following this cleanup, all chains are merged and renumbered starting from zero in a seamless way, with any discontinuous sites noted for subsequent procedures. Subsequently, the structure is iterated through, commencing from the first residue and bypassing any discontinued sites. During this process, fragments ranging in length from one to eight residues are systematically isolated. The surrounding residues housing at least one heavy atom within a 6Å proximity of the fragments are designated as the pocket—excluding those that fall within a 5-residue range in the protein sequence.

The derived complexes are subjected to stratified sampling based on the PDBBind2019 distribution. Pocket sizes, quantified by residue numbers, and ligand sizes, computed as effective residue numbers via molecular weight divided by 110Da, are key metrics for this stratification. Pseudo-complexes are sampled to approximate the joint distribution of ligand and pocket sizes. Another critical aspect considered is the relative buried surface area (rBSA), computed using the FreeSASA package in Python. Our pseudo-complexes exhibit two notable disparities from PDBBind2020 complexes after aligning the joint distribution of pocket sizes and ligand sizes in terms of rBSA. Primarily, the maximum rBSA of our complexes largely remains below 0.9 due to the removal of the nearest 5 residues at both ends of the protein sequence, creating two voids. Furthermore, there is an increased frequency of pairs with rBSA around 0.8. Although the first difference is inherent to the data synthesis process, the second one is addressed through targeted downsampling. Ultimately, to closely mimic native proteins, acetylation is applied to the N terminal, while amidation is applied to the C terminal. Following these comprehensive processing steps, our dataset comprises approximately 5.5 million ligand-pocket pairs.

A simplified algorithm is presented in Algorithm 1 to illustrate key steps.

---

**Algorithm 1** The Construction of Pseudo-Ligand-Pocket Complexes

---

**Require:** $R = [r_i]$ is an array of residues (a protein chain); $N$ is the largest peptide size; $thres$ is the distance threshold to define pockets; $p(i, j)$ is a group of pre-calculated sampling rates given a peptide size of $i$ and pocket size of $j$; $d(\cdot, \cdot)$ is a function measuring shortest distance between two residues or residue arrays.
1: **for** $r_i$ in $R$ **do**
2:    **for** $j = 1$ to $N$ **do**
3:       $lig \leftarrow [r_k$ for $k = i$ to $i + j - 1]$
4:       $poc \leftarrow [r_s$ if $d(lig, r_s) < thres$ for $r_s$ in $R]$
5:       **if** $rand() < p(j, |poc|)$ **then**
6:          **yield** $(lig, poc)$
7:       **end if**
8:    **end for**
9: **end for**

---

# B TRANSFERRING FROM PSEUDO-LIGANDS TO TRUE LIGANDS

In this section, we show how a good encoder can help generalize the learned contrastive representation from the pseudo-ligands to the true ligands. For the convenience of reading, we restate the formulations of training loss here. $g_T$ and $g_S$ are the predictor networks for protein and ligand respectively. The contrastive loss is given by

$$\mathcal{L}_{CL} = E_{p(\boldsymbol{t},\boldsymbol{s})}[\mathcal{L}_1(\boldsymbol{t}, \boldsymbol{s}) + \mathcal{L}_2(\boldsymbol{t}, \boldsymbol{s})] \tag{4}$$

$$\mathcal{L}_1(\boldsymbol{t}, \boldsymbol{s}) = -\log \frac{e^{g_T(\boldsymbol{t}) \cdot g_S(\boldsymbol{s})}}{\sum_{\boldsymbol{s}_i \in \mathbb{N}_s \cup \{\boldsymbol{s}\}} e^{g_T(\boldsymbol{t}) \cdot g_S(\boldsymbol{s}_i)}} = -g_T(\boldsymbol{t}) \cdot g_S(\boldsymbol{s}) + \log \sum_{\boldsymbol{s}_i \in \mathbb{N}_s \cup \{\boldsymbol{s}\}} e^{g_T(\boldsymbol{t}) \cdot g_S(\boldsymbol{s}_i)} \tag{5}$$

$$\mathcal{L}_2(\boldsymbol{t}, \boldsymbol{s}) = -\log \frac{e^{g_T(\boldsymbol{t}) \cdot g_S(\boldsymbol{s})}}{\sum_{\boldsymbol{t}_i \in \mathbb{N}_t \cup \{\boldsymbol{t}\}} e^{g_T(\boldsymbol{t}_i) \cdot g_S(\boldsymbol{s})}} = -g_T(\boldsymbol{t}) \cdot g_S(\boldsymbol{s}) + \log \sum_{\boldsymbol{t}_i \in \mathbb{N}_t \cup \{\boldsymbol{t}\}} e^{g_T(\boldsymbol{t}_i) \cdot g_S(\boldsymbol{s})}, \quad (6)$$

where $(\boldsymbol{t}, \boldsymbol{s})$ is a positive pair, $\mathbb{N}_s$ and $\mathbb{N}_t$ are in batch negative samples of protein pockets and pseudo-ligands, respectively.

For any protein pocket $p$ and its corresponding pseudo-ligand $l$, suppose there exists a potential real ligand $l^{(0)}$, denote their encoding vectors as $\boldsymbol{s}, \boldsymbol{t}, \boldsymbol{t}^{(0)}$.

**Theorem B.1.** *Assume $g_T$ is Lipschitz continuous with Lipschitz constant $l_T$ and $g_S$ is normalized. $\exists C_1, C_2$ are constants, $\forall g_T(\boldsymbol{t}^{(m)})$ in the line segment with $g_T(\boldsymbol{t})$ and $g_T(\boldsymbol{t}^{(0)})$ as endpoints, $\mathcal{L}_1(\boldsymbol{t}^{(m)}, \boldsymbol{s}) \leq C_1$, $\mathcal{L}_2(\boldsymbol{t}^{(m)}, \boldsymbol{s}) \leq C_2$, The pseudo-ligand's encoding is close to the true ligand's encoding:*

$$||\boldsymbol{t}_j - \boldsymbol{t}_j^{(0)}|| < \frac{1}{2l_T}, \ \forall \boldsymbol{t}_j \in \mathbb{N}_t \cup \{\boldsymbol{t}\}, \ \boldsymbol{t}_j^{(0)} \in \mathbb{N}_t^{(0)} \cup \{\boldsymbol{t}^{(0)}\}. \quad (7)$$

*Then $\forall \epsilon > 0$, when the pre-training loss is sufficiently small: $\mathcal{L}_1(\boldsymbol{t}, \boldsymbol{s}) < \sup_{\alpha \in (0,1)} \frac{\alpha \epsilon}{\lceil ln \frac{(1-\alpha)\epsilon}{C_1} / ln M_1 \rceil}$, $\mathcal{L}_2(\boldsymbol{t}, \boldsymbol{s}) < \sup_{\beta \in (0,1)} \frac{\beta \epsilon}{\lceil ln \frac{(1-\beta)\epsilon}{C_2} / ln M_2 \rceil}$, the contrastive loss concerning the pocket and the real ligand can be arbitrarily small: $\mathcal{L}_1(\boldsymbol{t}^{(0)}, \boldsymbol{s}) \leq \epsilon$, $\mathcal{L}_2(\boldsymbol{t}^{(0)}, \boldsymbol{s}) \leq \epsilon$.*

$M_1$, $M_2$ are constants defined in the proof. Before proving the theorem, we provide a helpful lemma with its proof.

**Lemma B.2.** *Assume $g_T$ is Lipschitz continuous with Lipschitz constant $l_T$ and $g_S$ is normalized. When equation 7 holds, we have*

$$\mu_1(\boldsymbol{t}, \boldsymbol{s}, \boldsymbol{t}^{(0)}, \boldsymbol{s}_i) \triangleq |(g_T(\boldsymbol{t}) - g_T(\boldsymbol{t}^{(0)}))(g_S(\boldsymbol{s}) - g_S(\boldsymbol{s}_i))| < 1, \ \forall \boldsymbol{s}_i \in \mathbb{N}_s, \quad (8)$$

$$\mu_2(\boldsymbol{t}, \boldsymbol{s}, \boldsymbol{t}^{(0)}, \boldsymbol{t}_j, \boldsymbol{t}_j^{(0)}) \triangleq |g_S(\boldsymbol{s})[(g_T(\boldsymbol{t}) - g_T(\boldsymbol{t}^{(0)})) - (g_T(\boldsymbol{t}_j) - g_T(\boldsymbol{t}_j^{(0)}))]| < 1, \ \forall \boldsymbol{t}_j \in \mathbb{N}_t, \ \boldsymbol{t}_j^{(0)} \in \mathbb{N}_t^{(0)}. \quad (9)$$

*Proof.* By definition of Lipschitz continuous, $||g_T(\boldsymbol{t}_j) - g_T(\boldsymbol{t}_j^{(0)})|| \leq l_T ||\boldsymbol{t}_j - \boldsymbol{t}_j^{(0)}|| < \frac{1}{2}$. According to Cauchy-Schwarz inequality, $M_1 = |(g_T(\boldsymbol{t}) - g_T(\boldsymbol{t}^{(0)}))(g_S(\boldsymbol{s}) - g_S(\boldsymbol{s}_i))| \leq ||(g_T(\boldsymbol{t}) - g_T(\boldsymbol{t}^{(0)}))|| ||(g_S(\boldsymbol{s}) - g_S(\boldsymbol{s}_i))|| < \frac{1}{2} \times 2 = 1$; $M_2 = |g_S(\boldsymbol{s})[(g_T(\boldsymbol{t}) - g_T(\boldsymbol{t}^{(0)})) - (g_T(\boldsymbol{t}_j) - g_T(\boldsymbol{t}_j^{(0)}))]| \leq ||g_S(\boldsymbol{s})|| ||(g_T(\boldsymbol{t}) - g_T(\boldsymbol{t}^{(0)})) - (g_T(\boldsymbol{t}_j) - g_T(\boldsymbol{t}_j^{(0)}))|| \leq ||g_S(\boldsymbol{s})|| (||(g_T(\boldsymbol{t}) - g_T(\boldsymbol{t}^{(0)}))|| + ||(g_T(\boldsymbol{t}_j) - g_T(\boldsymbol{t}_j^{(0)}))||) < 1.$ $\square$

*Proof of theorem B.1.* Our goal is to estimate the generalization gap, i.e. the loss gap when substituting the pseudo-ligand with the true ligand:

$$L^{(0)} - L = E_{p(\boldsymbol{t}^{(0)}, \boldsymbol{t}, \boldsymbol{s})}[(\mathcal{L}_1(\boldsymbol{t}^{(0)}, \boldsymbol{s}) - \mathcal{L}_1(\boldsymbol{t}, \boldsymbol{s})) + (\mathcal{L}_2(\boldsymbol{t}^{(0)}, \boldsymbol{s}) - \mathcal{L}_2(\boldsymbol{t}, \boldsymbol{s}))], \quad (10)$$

For the first term in equation 10:

$$\mathcal{L}_1(\boldsymbol{t}^{(0)}, \boldsymbol{s}) - \mathcal{L}_1(\boldsymbol{t}, \boldsymbol{s}) = \left( g_T(\boldsymbol{t}) \cdot g_S(\boldsymbol{s}) - g_T(\boldsymbol{t}^{(0)}) \cdot g_S(\boldsymbol{s}) \right)$$
$$- \left( \log \sum_{\boldsymbol{s}_i \in \mathbb{N}_s \cup \{\boldsymbol{s}\}} e^{g_T(\boldsymbol{t}) \cdot g_S(\boldsymbol{s}_i)} - \log \sum_{\boldsymbol{s}_i \in \mathbb{N}_s \cup \{\boldsymbol{s}\}} e^{g_T(\boldsymbol{t}^{(0)}) \cdot g_S(\boldsymbol{s}_i)} \right) \quad (11)$$

It can be simplified by mean value theorem as follows. $\exists g_T(\boldsymbol{t}^{(1)})$ in the line segment with $g_T(\boldsymbol{t})$ and $g_T(\boldsymbol{t}^{(0)})$ as endpoints, such that equation 11

$$= (g_T(\boldsymbol{t}) - g_T(\boldsymbol{t}^{(0)})) \cdot g_S(\boldsymbol{s}) - \left( \frac{\sum_{\boldsymbol{s}_i \in \mathbb{N}_s \cup \{\boldsymbol{s}\}} e^{g_T(\boldsymbol{t}^{(1)}) \cdot g_S(\boldsymbol{s}_i)} g_S(\boldsymbol{s}_i)}{\sum_{\boldsymbol{s}_i \in \mathbb{N}_s \cup \{\boldsymbol{s}\}} e^{g_T(\boldsymbol{t}^{(1)}) \cdot g_S(\boldsymbol{s}_i)}} (g_T(\boldsymbol{t}) - g_T(\boldsymbol{t}^{(0)})) \right) \quad (12)$$

Denote $\lambda_1(\boldsymbol{s}_i, \boldsymbol{t}^{(1)}) \triangleq \frac{e^{g_T(\boldsymbol{t}^{(1)}) \cdot g_S(\boldsymbol{s}_i)}}{\sum_{\boldsymbol{s}_i \in \mathbb{N}_s \cup \{\boldsymbol{s}\}} e^{g_T(\boldsymbol{t}^{(1)}) \cdot g_S(\boldsymbol{s}_i)}} \in [0, 1]$, They satisfy $\sum_{\boldsymbol{s}_i \in \mathbb{N}_s \cup \{\boldsymbol{s}\}} \lambda_1(\boldsymbol{s}_i, \boldsymbol{t}^{(1)}) = 1$. Then equation 12

$$= (g_T(\boldsymbol{t}) - g_T(\boldsymbol{t}^{(0)})) \cdot g_S(\boldsymbol{s}) - \left( \sum_{\boldsymbol{s}_i \in \mathbb{N}_s \cup \{\boldsymbol{s}\}} \lambda_1(\boldsymbol{s}_i, \boldsymbol{t}^{(1)}) g_S(\boldsymbol{s}_i)(g_T(\boldsymbol{t}) - g_T(\boldsymbol{t}^{(0)})) \right) \quad (13)$$

$$= (g_T(\boldsymbol{t}) - g_T(\boldsymbol{t}^{(0)})) \left( g_S(\boldsymbol{s}) - \sum_{\boldsymbol{s}_i \in \mathbb{N}_s \cup \{\boldsymbol{s}\}} \lambda_1(\boldsymbol{s}_i, \boldsymbol{t}^{(1)}) g_S(\boldsymbol{s}_i) \right) \quad (14)$$

$$= (g_T(\boldsymbol{t}) - g_T(\boldsymbol{t}^{(0)})) \left( \sum_{\boldsymbol{s}_i \in \mathbb{N}_s \cup \{\boldsymbol{s}\}} \lambda_1(\boldsymbol{s}_i, \boldsymbol{t}^{(1)})(g_S(\boldsymbol{s}) - g_S(\boldsymbol{s}_i)) \right) \quad (15)$$

$$= (g_T(\boldsymbol{t}) - g_T(\boldsymbol{t}^{(0)})) \left( \sum_{\boldsymbol{s}_i \in \mathbb{N}_s} \lambda_1(\boldsymbol{s}_i, \boldsymbol{t}^{(1)})(g_S(\boldsymbol{s}) - g_S(\boldsymbol{s}_i)) \right) \quad (16)$$

Note that $\sum_{\boldsymbol{s}_i \in \mathbb{N}_s} \lambda_1(\boldsymbol{s}_i, \boldsymbol{t}^{(1)}) = 1 - \frac{e^{g_T(\boldsymbol{t}^{(1)}) \cdot g_S(\boldsymbol{s})}}{\sum_{\boldsymbol{s}_i \in \mathbb{N}_s \cup \{\boldsymbol{s}\}} e^{g_T(\boldsymbol{t}^{(1)}) \cdot g_S(\boldsymbol{s}_i)}} = 1 - exp(-\mathcal{L}_1(\boldsymbol{t}^{(1)}, \boldsymbol{s})) \le \mathcal{L}_1(\boldsymbol{t}^{(1)}, \boldsymbol{s})$, and by assumption $\forall \boldsymbol{s}_i \in \mathbb{N}_s, \mu_1 = |(g_T(\boldsymbol{t}) - g_T(\boldsymbol{t}^{(0)}))(g_S(\boldsymbol{s}) - g_S(\boldsymbol{s}_i))| < 1$, let $M_1 \triangleq \max_{\boldsymbol{s}_i \in \mathbb{N}_s} \mu_1(\boldsymbol{t}, \boldsymbol{s}, \boldsymbol{t}^{(0)}, \boldsymbol{s}_i)$, then $M_1 < 1$ and

$$|\mathcal{L}_1(\boldsymbol{t}^{(0)}, \boldsymbol{s}) - \mathcal{L}_1(\boldsymbol{t}, \boldsymbol{s})| \le M_1 \mathcal{L}_1(\boldsymbol{t}^{(1)}, \boldsymbol{s}). \quad (17)$$

Consequently, we obtain a recursive formula

$$\begin{aligned} \mathcal{L}_1(\boldsymbol{t}^{(0)}, \boldsymbol{s}) &\le |\mathcal{L}_1(\boldsymbol{t}^{(0)}, \boldsymbol{s}) - \mathcal{L}_1(\boldsymbol{t}, \boldsymbol{s})| + \mathcal{L}_1(\boldsymbol{t}, \boldsymbol{s}) \\ &\le M_1 \mathcal{L}_1(\boldsymbol{t}^{(1)}, \boldsymbol{s}) + \mathcal{L}_1(\boldsymbol{t}, \boldsymbol{s}) \\ &\le M_1^m \mathcal{L}_1(\boldsymbol{t}^{(m)}, \boldsymbol{s}) + m \mathcal{L}_1(\boldsymbol{t}, \boldsymbol{s}), \end{aligned} \quad (18)$$

equation 18 holds for any $m \in \mathbb{N}^+$. By assumption, $\exists \alpha \in (0, 1), \mathcal{L}_1(\boldsymbol{t}, \boldsymbol{s}) \le \frac{\alpha\epsilon}{\lceil ln\frac{(1-\alpha)\epsilon}{C_1}/lnM_1 \rceil}$, $\mathcal{L}_1(\boldsymbol{t}^{(m)}, \boldsymbol{s}) \le C_1$, then let $m = \lceil ln\frac{(1-\alpha)\epsilon}{C_1}/lnM_1 \rceil$, we obtain $\mathcal{L}_1(\boldsymbol{t}^{(0)}, \boldsymbol{s}) \le M_1^{ln\frac{(1-\alpha)\epsilon}{C_1}/lnM_1} C_1 + m\frac{\alpha\epsilon}{m} = \epsilon$.

Similar results can be proved for the second term in equation 10.

$$\mathcal{L}_2(\boldsymbol{t}^{(0)}, \boldsymbol{s}) - \mathcal{L}_2(\boldsymbol{t}, \boldsymbol{s}) = \left( g_T(\boldsymbol{t}) \cdot g_S(\boldsymbol{s}) - g_T(\boldsymbol{t}^{(0)}) \cdot g_S(\boldsymbol{s}) \right) -$$
$$\left( \log \sum_{\boldsymbol{t}_i \in \mathbb{N}_t \cup \{\boldsymbol{t}\}} e^{g_T(\boldsymbol{t}_i) \cdot g_S(\boldsymbol{s})} - \log \sum_{\boldsymbol{t}_i^{(0)} \in \mathbb{N}_t^{(0)} \cup \{\boldsymbol{t}^{(0)}\}} e^{g_T(\boldsymbol{t}_i^{(0)}) \cdot g_S(\boldsymbol{s})} \right). \quad (19)$$

To make the index clear, assume batch size is N, the positive triplet sets are $(\boldsymbol{s}_i, \boldsymbol{t}_i, \boldsymbol{t}_i^{(0)})_{i=0,\cdots,N-1}$ and denote $(\boldsymbol{s}_0, \boldsymbol{t}_0, \boldsymbol{t}_0^{(0)}) \triangleq (\boldsymbol{s}, \boldsymbol{t}, \boldsymbol{t}^{(0)})$ is the anchor in equation 19. Then we use the mean value theorem:$\forall i \in \{0, \cdots, N-1\}, \exists g_T(\boldsymbol{t}_i^{(1)})$ in the line segment with $g_T(\boldsymbol{t}_i)$ and $g_T(\boldsymbol{t}_i^0))$ as endpoints, such that equation 19

$$= (g_T(\boldsymbol{t}) - g_T(\boldsymbol{t}^{(0)})) \cdot g_S(\boldsymbol{s}) - \left( \sum_{j=0}^{N-1} \frac{e^{g_T(\boldsymbol{t}_j^{(1)}) \cdot g_S(\boldsymbol{s})} g_S(\boldsymbol{s})}{\sum_{i=0}^{N-1} e^{g_T(\boldsymbol{t}_i^{(1)}) \cdot g_S(\boldsymbol{s})}} (g_T(\boldsymbol{t}_j) - g_T(\boldsymbol{t}_j^{(0)})) \right) \quad (20)$$

Denote $\lambda_2(\boldsymbol{s}, \boldsymbol{t}_j^{(1)}) \triangleq \frac{e^{g_T(\boldsymbol{t}_j^{(1)}) \cdot g_S(\boldsymbol{s})}}{\sum_{i=0}^{N-1} e^{g_T(\boldsymbol{t}_i^{(1)}) \cdot g_S(\boldsymbol{s})}} \in [0, 1]$. They satisfy $\sum_{j=0}^{N-1} \lambda_2(\boldsymbol{s}, \boldsymbol{t}_j^{(1)}) = 1$. Then equation 20

$$= (g_T(\boldsymbol{t}) - g_T(\boldsymbol{t}^{(0)})) \cdot g_S(\boldsymbol{s}) - \left( \sum_{j=0}^{N-1} \lambda_2(\boldsymbol{s}, \boldsymbol{t}_j^{(1)}) g_S(\boldsymbol{s})(g_T(\boldsymbol{t}_j) - g_T(\boldsymbol{t}_j^{(0)})) \right) \tag{21}$$

$$= g_S(\boldsymbol{s}) \left( (g_T(\boldsymbol{t}) - g_T(\boldsymbol{t}^{(0)})) - \sum_{j=0}^{N-1} \lambda_2(\boldsymbol{s}, \boldsymbol{t}_j^{(1)})(g_T(\boldsymbol{t}_j) - g_T(\boldsymbol{t}_j^{(0)})) \right) \tag{22}$$

$$= g_S(\boldsymbol{s}) \left( \sum_{j=0}^{N-1} \lambda_2(\boldsymbol{s}, \boldsymbol{t}_j^{(1)})[(g_T(\boldsymbol{t}) - g_T(\boldsymbol{t}^{(0)})) - (g_T(\boldsymbol{t}_j) - g_T(\boldsymbol{t}_j^{(0)}))] \right) \tag{23}$$

$$= g_S(\boldsymbol{s}) \left( \sum_{j=1}^{N-1} \lambda_2(\boldsymbol{s}, \boldsymbol{t}_j^{(1)})[(g_T(\boldsymbol{t}) - g_T(\boldsymbol{t}^{(0)})) - (g_T(\boldsymbol{t}_j) - g_T(\boldsymbol{t}_j^{(0)}))] \right). \tag{24}$$

Note that $\sum_{j=1}^{N-1} \lambda_2(\boldsymbol{s}, \boldsymbol{t}_j^{(1)}) = 1 - \frac{e^{g_T(\boldsymbol{t}^{(1)}) \cdot g_S(\boldsymbol{s})}}{\sum_{i=0}^{N-1} e^{g_T(\boldsymbol{t}_i^{(1)}) \cdot g_S(\boldsymbol{s})}} = 1 - exp(-\mathcal{L}_2(\boldsymbol{t}^{(1)}, \boldsymbol{s})) \leq \mathcal{L}_2(\boldsymbol{t}^{(1)}, \boldsymbol{s})$, and by assumption $\forall j \in \{1, \cdots, N-1\}, \mu_2 = |g_S(\boldsymbol{s})[(g_T(\boldsymbol{t}) - g_T(\boldsymbol{t}^{(0)})) - (g_T(\boldsymbol{t}_j) - g_T(\boldsymbol{t}_j^{(0)}))]| < 1$, let $M_2 \triangleq \max_{j \in \{1, \cdots, N-1\}} \mu_2(\boldsymbol{t}, \boldsymbol{s}, \boldsymbol{t}^{(0)}, \boldsymbol{t}_j, \boldsymbol{t}_j^{(0)})$, then $M_2 < 1$ and

$$|\mathcal{L}_2(\boldsymbol{t}^{(0)}, \boldsymbol{s}) - \mathcal{L}_2(\boldsymbol{t}, \boldsymbol{s})| \leq M_2 \mathcal{L}_2(\boldsymbol{t}^{(1)}, \boldsymbol{s}). \tag{25}$$

Consequently, we obtain a recursive formula similar to equation 18:

$$\begin{aligned} \mathcal{L}_2(\boldsymbol{t}^{(0)}, \boldsymbol{s}) &\leq |\mathcal{L}_2(\boldsymbol{t}^{(0)}, \boldsymbol{s}) - \mathcal{L}_2(\boldsymbol{t}, \boldsymbol{s})| + \mathcal{L}_2(\boldsymbol{t}, \boldsymbol{s}) \\ &\leq M_2 \mathcal{L}_2(\boldsymbol{t}^{(1)}, \boldsymbol{s}) + \mathcal{L}_2(\boldsymbol{t}, \boldsymbol{s}) \\ &\leq M_2^m \mathcal{L}_2(\boldsymbol{t}^{(m)}, \boldsymbol{s}) + m\mathcal{L}_2(\boldsymbol{t}, \boldsymbol{s}) \end{aligned} \tag{26}$$

equation 26 holds for any $m \in \mathbb{N}^+$. By assumption $\exists \beta \in (0, 1), \mathcal{L}_2(\boldsymbol{t}, \boldsymbol{s}) \leq \frac{\beta \epsilon}{\lceil ln \frac{(1-\beta)\epsilon}{C_2} / ln M_2 \rceil}$, $\mathcal{L}_2(\boldsymbol{t}^{(m)}, \boldsymbol{s}) \leq C_2$, then let $m = \lceil ln \frac{(1-\beta)\epsilon}{C_2} / ln M_2 \rceil$, we obtain $\mathcal{L}_2(\boldsymbol{t}^{(0)}, \boldsymbol{s}) \leq M_2^{ln \frac{(1-\beta)\epsilon}{C_2} / ln M_2} C_2 + m \frac{\beta \epsilon}{m} = \epsilon$. $\square$

An illustration of the result above is shown in Figuire 6a. We also visualize the overlapping between distributions of real ligands and protein fragments in Figuire 6b.

## C ADDITIONAL EXPERIMENT RESULTS

### C.1 FULL LBA PREDICTION RESULTS INCLUDE STANDARD DEVIATION

Results on LBA prediction task with standard deviation are listed in Table 6 and Table 7.

### C.2 ABLATION RESULTS ON LBA TASK

Results on LBA prediction with different pretrained models are listed in Table 8

### C.3 ABLATION STUDY ON FIXING MOLECULE ENCODER

The result is shown in Table 9.

### C.4 ABLATION STUDY ON DIFFERENT DISTANCE THRESHOLDS FOR POCKETS

The result is shown in Table 10

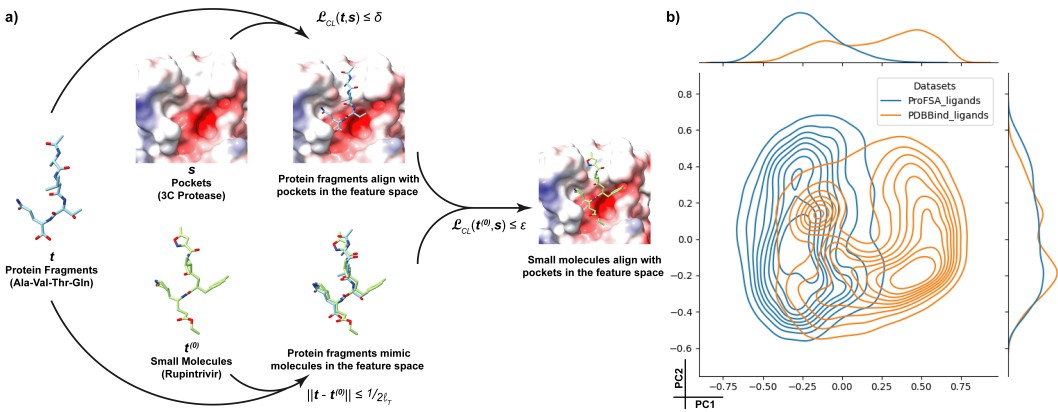

Figure 6: **a)** An illustration of Theorem B.1 that training with protein fragments could generalize to real ligands; **b)** The distributional overlapping between Uni-Mol-encoded real ligands and Uni-Mol-encoded protein fragments.

Table 6: Results of sequence identity 30% on LBA prediction task with standard deviation.

|  | **Method** | **Sequence Identity 30%** | | |
|---|---|---|---|---|
|  |  | RMSE $\downarrow$ | Pearson $\uparrow$ | Spearman $\uparrow$ |
| Sequence Based | DeepDTA | $1.866 \pm 0.08$ | $0.472 \pm 0.02$ | $0.471 \pm 0.02$ |
|  | B&B | $1.985 \pm 0.01$ | $0.165 \pm 0.01$ | $0.152 \pm 0.02$ |
|  | TAPE | $1.890 \pm 0.04$ | $0.338 \pm 0.04$ | $0.286 \pm 0.12$ |
|  | ProtTrans | $1.544 \pm 0.02$ | $0.438 \pm 0.05$ | $0.434 \pm 0.06$ |
| W/o Pretrain | Holoprot | $1.464 \pm 0.01$ | $0.509 \pm 0.00$ | $0.500 \pm 0.01$ |
|  | IEConv | $1.554 \pm 0.02$ | $0.414 \pm 0.05$ | $0.428 \pm 0.03$ |
|  | MaSIF | $1.484 \pm 0.02$ | $0.467 \pm 0.02$ | $0.455 \pm 0.01$ |
|  | 3DCNN | $1.416 \pm 0.02$ | $0.550 \pm 0.02$ | $0.553 \pm 0.01$ |
|  | ENN | $1.568 \pm 0.01$ | $0.389 \pm 0.02$ | $0.408 \pm 0.02$ |
|  | GNN | $1.601 \pm 0.05$ | $0.545 \pm 0.03$ | $0.533 \pm 0.03$ |
|  | ProNet | $1.463 \pm 0.00$ | $0.551 \pm 0.01$ | $0.551 \pm 0.01$ |
| With Pretrain | GeoSSL | $1.451 \pm 0.03$ | $0.577 \pm 0.02$ | $0.572 \pm 0.01$ |
|  | DeepAffinity | $1.893 \pm 0.65$ | $0.415$ | $0.426$ |
|  | EGNN-PLM | $1.403 \pm 0.01$ | $0.565 \pm 0.02$ | $0.544 \pm 0.01$ |
|  | Uni-Mol | $1.520 \pm 0.03$ | $0.558 \pm 0.00$ | $0.540 \pm 0.00$ |
|  | ProFSA | $1.377 \pm 0.01$ | $0.628 \pm 0.01$ | $0.620 \pm 0.01$ |

## C.5 EXPERIMENT RESULTS OF LIGAND EFFICACY PREDICTION

The result is shown in Table 11. We follow the similar setting used in ATOM3D (Townshend et al., 2020) .

## C.6 EXPERIMENT RESULTS OF PROTEIN-PROTEIN AFFINITY PREDICTION

The result is shown in Table 12. Baseline results are from GET (Anonymous, 2023) . Briefly, the flexible PPI dataset is acquired from the anonymous code base of the GET. Protein complexes and binding-free monomers are superposed, and unaligned residues are removed unless they are on the interaction surface. Fragments of sizes from 1 to 7 are all isolated, and so are their corresponding pockets. The ProFSA contrastive scores are independently aggregated for receptors, ligands, and complexes. Interactions are approximated by the score of the complex minus the sore of the receptor and the ligand.

Table 7: Results of sequence identity 60% LBA prediction task with standard deviation.

| Method | | Sequence Identity 60% | | |
|---|---|---|---|---|
| | | RMSE ↓ | Pearson ↑ | Spearman ↑ |
| Sequence Based | DeepDTA | $1.762 \pm 0.26$ | $0.666 \pm 0.01$ | $0.663 \pm 0.02$ |
| | B&B | $1.891 \pm 0.00$ | $0.249 \pm 0.01$ | $0.275 \pm 0.01$ |
| | TAPE | $1.633 \pm 0.02$ | $0.568 \pm 0.03$ | $0.571 \pm 0.02$ |
| | ProtTrans | $1.641 \pm 0.02$ | $0.595 \pm 0.01$ | $0.588 \pm 0.01$ |
| W/o Pretrain | Holoprot | $1.365 \pm 0.04$ | $0.749 \pm 0.01$ | $0.742 \pm 0.01$ |
| | IEConv | $1.473 \pm 0.02$ | $0.667 \pm 0.01$ | $0.675 \pm 0.02$ |
| | MaSIF | $1.426 \pm 0.02$ | $0.709 \pm 0.01$ | $0.701 \pm 0.00$ |
| | 3DCNN | $1.621 \pm 0.03$ | $0.608 \pm 0.02$ | $0.615 \pm 0.03$ |
| | ENN | $1.620 \pm 0.05$ | $0.623 \pm 0.02$ | $0.633 \pm 0.02$ |
| | GNN | $1.408 \pm 0.07$ | $0.743 \pm 0.02$ | $0.743 \pm 0.03$ |
| | ProNet | $1.343 \pm 0.03$ | $0.765 \pm 0.01$ | $0.761 \pm 0.00$ |
| With Pretrain | EGNN-PLM | $1.559 \pm 0.02$ | $0.644 \pm 0.02$ | $0.646 \pm 0.02$ |
| | Uni-Mol | $1.619 \pm 0.04$ | $0.645 \pm 0.02$ | $0.653 \pm 0.02$ |
| | ProFSA | $1.377 \pm 0.01$ | $0.764 \pm 0.00$ | $0.762 \pm 0.01$ |

Table 8: Results on LBA prediction task with different model setting.

| Method | Sequence Identity 30% | | | Sequence Identity 60% | | |
|---|---|---|---|---|---|---|
| | RMSE ↓ | Pearson ↑ | Spearman ↑ | RMSE ↓ | Pearson ↑ | Spearman ↑ |
| ProFSA-Uni-Mol | 1.353 | 0.628 | 0.623 | 1.346 | 0.764 | 0.762 |
| ProFSA-Frad | 1.382 | 0.606 | 0.587 | 1.289 | 0.784 | 0.777 |
| ProFSA-PTVD | 1.412 | 0.609 | 0.593 | 1.354 | 0.762 | 0.753 |
| ProFSA-no-align | 1.441 | 0.570 | 0.540 | 1.429 | 0.732 | 0.727 |
| Uni-mol | 1.520 | 0.558 | 0.540 | 1.619 | 0.645 | 0.653 |

## D  ENCODER ARCHITECTURE

First, each protein pocket is tokenized into its individual atoms. A pocket with $L$ tokens is described by the feature vector $x^p = \{c_p, t_p\}$, where $c_p \in \mathbb{R}^{L \times 3}$ represents the 3D coordinates of the atoms and $t_p \in \mathbb{R}^L$ denotes the types of atoms present.

The core component of our encoder is an invariant 3D transformer, tailored for ingesting these tokenized atom features. Initially, a pairwise representation, denoted by $q_{ij}^0$, is computed based on the distances between each pair of atoms.

For each layer $l$ in the transformer, the self-attention mechanism for learning atom-level representations is formulated as specified in Equation 27. Importantly, this pairwise representation serves as a bias term in the attention mechanism, enabling the encoding of 3D spatial features into the atom representations. The rules for updating between adjacent transformer layers are also defined in Equation 27.

$$\text{Attention}(Q_i^l, K_j^l, V_j^l) = \text{softmax}(\frac{Q_i^l (K_j^l)^T}{\sqrt{d}} + q_{ij}^l) V_j^l, \quad \text{where } q_{ij}^{l+1} = q_{ij}^l + \frac{Q_i^l (K_j^l)^T}{\sqrt{d}}. \quad (27)$$

## E  EXPERIMENT DETAILS

### E.1  PRETRAINING

During the pretraining phase, we utilize a batch size of $4 \times 48$ on 4 Nvidia A100 GPUs. We choose the Adam optimizer with a learning rate of $1 \times 10^{-4}$ and cap the training at 100 epochs. A polynomial decay scheduler with a warmup ratio of 0.06 is implemented. The checkpoint yielding the best validation AUC is retained, complemented by an early stopping strategy set with a 20-epoch patience. The training lasts approximately 10 days.

Table 9: Impact of fixing molecule encoder.

| | Druggability Score (Zero-Shot) ↓ | | | | Pocket Matching ↑ | |
| | Fpocket | Druggability | Total Sasa | Hydrophobicity | Kahraman(w/o $PO_4$) | Tough M1 |
|---|---|---|---|---|---|---|
| ProFSA | 0.1238 | 0.1090 | 31.17 | 12.01 | 0.7870 | 0.8178 |
| w/o fixed mol encoder | 0.1247 | 0.1094 | 32.17 | 12.20 | 0.6905 | 0.7337 |

Table 10: Impact of different distance thresholds.

| | Druggability Score (Zero-Shot) ↓ | | | | Pocket Matching ↑ | |
| | Fpocket | Druggability | Total Sasa | Hydrophobicity | Kahraman(w/o $PO_4$) | Tough M1 |
|---|---|---|---|---|---|---|
| 4Å | 0.1240 | 0.1095 | 28.29 | 13.07 | 0.7062 | 0.7549 |
| 6Å | 0.1238 | 0.1090 | 31.17 | 12.01 | 0.7870 | 0.8178 |
| 8Å | 0.1256 | 0.1125 | 34.83 | 12.92 | 0.8322 | 0.8292 |

### E.2 DRUGGABILITY PREDICTION

**Dataset Description**  We employ the regression dataset created by Uni-Mol (Zhou et al., 2023) to evaluate the pocket druggability prediction performance of ProFSA. The dataset comprises 164,586 candidate pockets and four scores: Fpocket Score, Druggability Score, Total SASA, and Hydrophobicity Score. These scores are calculated using Fpocket (Guilloux et al., 2009) and indicate the druggability of candidate pockets.

**Finetuning Setting**  We finetune our model by integrating a Multiple Layer Perceptron (MLP) with one hidden layer of 512 dimensions for regression. Utilizing 4 Nvidia A100 GPUs, we adopt a batch size of $32 \times 4$. We employ the Adam optimizer with a learning rate of $1 \times 10^{-4}$. The training spans up to 200 epochs. We use a polynomial decay scheduler with a warmup ratio of 0.03. We select the checkpoint that delivers the lowest validation RMSE and incorporate an early stopping mechanism with a patience of 20 epochs. For the Hydrophobicity score, we set the max epoch to 500 and the learning rate to $1 \times 10^{-3}$.

**zero-shot setting**  We employ KNN regression, an unsupervised technique, to estimate the score. For an encoded pocket $t$ from the test set, we identify its top $k$ closest "neighbors" in the training set, represented as $t_1, t_2, ..., t_k$ with associated labels $l_1, l_2, ..., l_k$. The similarity between the representations is gauged using the cosine similarity defined as:

$$s_i = \frac{t \cdot t_i}{\|t\| \cdot \|t_i\|}$$

The predicted score for pocket $t$ is a weighted average of the labels where weights are inversely proportional to their similarity.

$$\hat{l}(t) = \frac{\sum_{i=1}^{k} \left( l_i \cdot \frac{1}{s_i + \epsilon} \right)}{\sum_{i=1}^{k} \left( \frac{1}{s_i + \epsilon} \right)} \tag{28}$$

Here we use $k = 200$.

### E.3 POCKET MATCHING

**Dataset Description**  We have given an explicit introduction of two datasets utilized for pocket matching task, please refer to Section 4.2 for detailed information.

**zero-shot Setting**  For the zero-shot setting, we use the cosine similarity of encoded embeddings to define whether two pockets are similar. For the TOUGH-M1 dataset, the pocket is defined as the surrounding residues of the real ligands housing at least one heavy atom within an 8Åproximity of the fragments.

Table 11: Result of Ligand Efficacy Prediction. The best result is highlighted in **bold**.

| | AUROC ↑ | AUPRC ↑ |
|---|---|---|
| ATOM3D-GNN | 0.681 | 0.598 |
| GeoSSL | 0.776±0.03 | 0.694±0.06 |
| Uni-Mol | 0.782±0.02 | 0.695±0.07 |
| ProFSA | **0.840±0.04** | **0.806±0.04** |

Table 12: Result of Protein-Protein Affinity Prediction in the flexible setting. Results ranking first and second are highlighted in **bold** and underlined respectively.

| | Spearman ↑ |
|---|---|
| SchNet | 0.072 ± 0.021 |
| DimeNet++ | 0.171 ± 0.054 |
| EGNN | 0.080 ± 0.038 |
| TorchMD | 0.117 ± 0.008 |
| GET | **0.363 ± 0.017** |
| ProFSA(zero-shot) | 0.248 |

**Finetuning Setting**   We use the same data and similar loss provided by Simonovsky & Meyers (2020). Specifically, the loss is defined as

$$l = -y * p - (1 - y) * (1 - p) \tag{29}$$

where $p = \frac{t \cdot t_i}{\|t\| \cdot \|t_i\|}$

For the TOUGH-M1 test set, we employ a batch size of $64 \times 4$ across 4 Nvidia A100 GPUs. The training is limited to a maximum of 20 epochs using the Adam optimizer with a learning rate of $1 \times 10^{-5}$. We use a polynomial decay scheduler with a warmup ratio of 0.1.

For the Kahraman dataset, the batch size is set to $64 \times 4$. Training process for up to 5 epochs using the Adam optimizer at a learning rate of $3 \times 10^{-7}$. We use a polynomial decay scheduler with a warmup ratio of 0.2.

### E.4   POCKET-LIGAND BINDING AFFINITY PREDICTION

**Dataset Description**   We use the dataset curated from PDBBind(v2019) and experiment settings in Atom3D (Townshend et al., 2022), adopting dataset split with 30% and 60% sequence identity thresholds to verify the generalization ability of ProFSA. As for the split based on a 30% sequence identity threshold, the resulting sizes of training, validation, and test sets are 3507, 466, and 490, respectively, while the 60% sequence identity threshold leads to counterparts of size 3678, 460, and 460, respectively.

**Finetuning Setting**   We refine our model by incorporating a Multiple Layer Perceptron (MLP) for regression, featuring four hidden layers with dimensions of 1024, 512, 256, and 128. Using 4 Nvidia A100 GPUs, we set the batch size to $16 \times 4$. The training leverages the Adam optimizer with a learning rate of $1 \times 10^{-4}$ and is conducted over a maximum of 50 epochs. A polynomial decay scheduler with a 0.2 warmup ratio is applied. The checkpoint with the optimal validation RMSE is chosen, and we implement an early stopping strategy with a patience threshold of 20 epochs.

During training, the parameters of both pocket encoders and molecule encoders are fixed.

## F   BIOLOGICAL JUSTIFICATIONS AND VISUALIZATIONS

Figure 7 illustrates that the pseudo pairs we created share various interaction types commonly found in real pocket-ligand pairs, such as hydrogen bonding, $\pi - \pi$ stacking, and salt bridge. In the

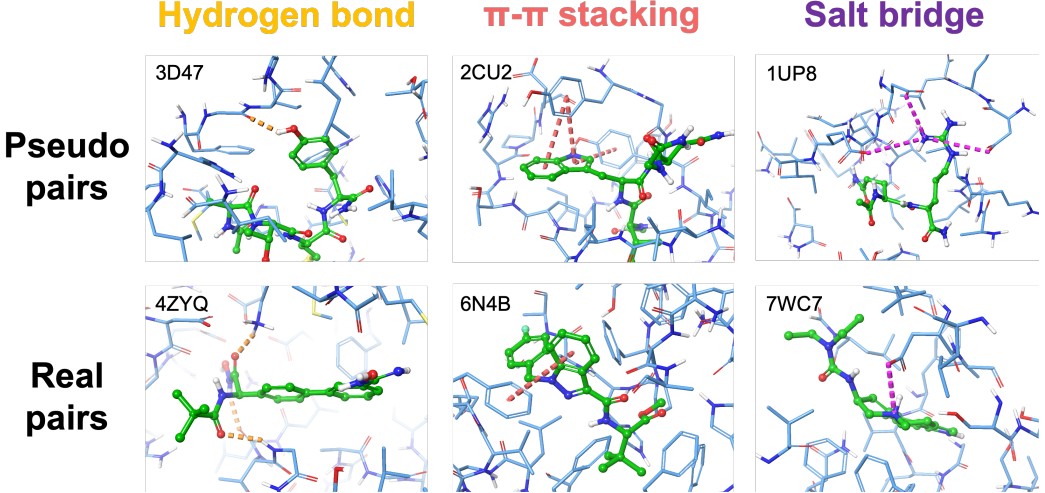

Figure 7: Visualization of binding patterns

figures, each type of interaction is represented by dashed lines, color-coded to correspond with the specific interaction type. This observation aligns with the findings reported by Polizzi & DeGrado (2020), which demonstrate that intra-protein interactions bear a strong resemblance to protein-ligand interactions. The presence of these similar interaction patterns substantiates our strategy of using peptides to mimic the behavior of small molecules. This approach is a key factor in explaining the success we observed in our benchmarks.

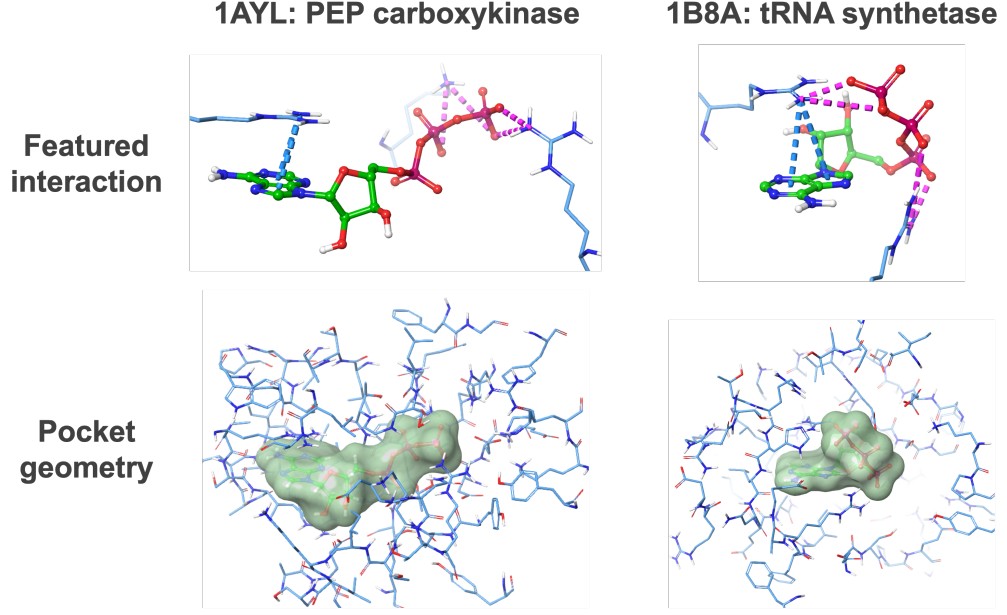

Figure 8: Visualization of similar pocket in Kahraman dataset

In Figure 8, we showed an example of two non-homology ATP binding pockets to explain why pocket-matching can benefit from interaction-aware pretraining so that we have achieved superior results in the Kahraman dataset and the BioLip t-SNE visualization. The PEP carboxykinase(PDB:1AYL) and tRNA synthetase(PDB:1B8A) are two ATP-binding proteins that share zero sequence similarity (verified with the BLAST) extracted from the Kahraman dataset. However, as they are both fueled by ATP, their binding site shares similar binding patterns. The cation-π in-

teractions(blue dash lines) and salt bridges (magenta dash lines) are important to the ATP binding, which can be viewed as convergent evolutions at the molecule level. Even though, these two pockets are very distinct in terms of shapes and sizes because they bind ATPs in different conformations. Therefore, biochemical interactions are the key to accomplishing the pocket-matching task, which is ignored in previous self-supervised learning methods like Uni-Mol.

# G    VISUALIZATION OF DISTRIBUTIONS OF DIFFERENT MOLECULAR PROPERTIES

Result is shown in Figure 9.

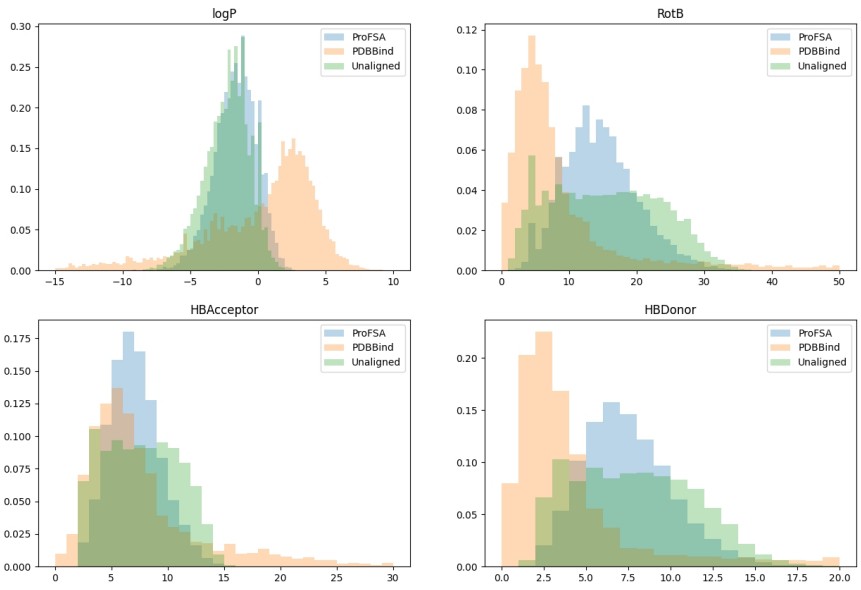

Figure 9: Visualization of distributions of different molecular properties

# H    VISUALIZATION OF C AND N TERMINALS

The result is shown in Figure 10. When a peptide bond is formed, one amine group and one carboxy group are left untouched. Therefore, in a linear polypeptide, there would always be a free amine group and a free carboxy group, and they are called the N terminal and the C terminal respectively in biochemistry.

Figure 10: Visualization of N terminal and C terminal