# OpenReview forum: "Self-supervised Pocket Pretraining via Protein Fragment-Surroundings Alignment"
_ICLR.cc/2024/Conference — ICLR 2024 poster_

### Official Review · Reviewer_bxUv · 2023-10-31

**Soundness:** 3 good
**Presentation:** 2 fair
**Contribution:** 2 fair
**Rating:** 6
**Confidence:** 4

**Summary:**

The authors propose a novel self-supervised pretraining approach called ProFSA to learn effective pocket representations by leveraging protein-only data. The key idea is to extract pseudo-ligand-pocket pairs from proteins by segmenting structures into fragments and designating the surroundings as pockets.

**Strengths:**

1. The authors present a novel pairwise data synthesis pipeline by extracting pseudo-ligand-pocket pairs from protein data
2. The authors develop large-scale datasets and new pretraining methods to exploit the full potential of pocket representation learning, emphasizing the interactions between pockets and ligands.
3. ProFSA achieves significant performance gains in a variety of downstream tasks.

**Weaknesses:**

1. The evidence for the construction of the pseudo-ligand is not clear.
2. Ablation studies evaluating the impact of critical design choices like fragment sizes, distance thresholds for pockets would provide useful insights.
3. While terminal corrections are applied to address biases from breaking peptide bonds, the pseudo-ligands may still exhibit substantial discrepancies from real drug-like ligands.

**Questions:**

1. Why do the authors choose peptides to replace small molecules, and is this choice reliable? Have the authors considered other potential ways to further close the gap between pseudo-ligands and real ligands, either through data processing strategies or by fine-tuning on downstream tasks?
2. Section 3.1, second paragraph, line 4, what do the N Terminal and C Terminal refer to?
3. Why fixed the molecule encoder in contrastive learning, i.e., the encoder that encodes the pseudo-ligand.
4. Could ProFSA be extended to other tasks like protein-protein interaction prediction? How might the pipeline and contrastive approach need to be adapted?

---

> ### Author Response · Authors · 2023-11-17
>
> # Response 1/2
>
> ## Response to evidence for the construction of the pseudo-ligand and using peptides to replace small molecules
>
> As we discussed in the paper, pseudo-ligands share similar sizes as real ligands, and they also make similar non-covalent interactions with the pocket. To further support our proposal, we would like to cite another paper **"A defined structural unit enables de novo design of small-molecule–binding proteins"**[1], published in Science 2020, which supports that **intra-protein interactions are similar to protein-ligand interactions.** That is the reason we use peptides to **represent** small molecules to let the pocket encoder learn the interaction information.
>
> Also, we'd like to clarify that our intention is not to **"replace"** small molecules with peptides. It's a pretraining framework that leverages abundant protein-only data and uses peptides to **simulate** pocket-small molecule interactions for enhanced pocket representations. Following this pretraining phase, **the model can be finetuned with datasets that include real small molecules, ensuring its effectiveness in practical applications**.
>
> To demonstrate the validity of constructing pseudo-ligands, we provide visualizations in **Appendix F** showing that interactions between pockets and our pseudo-ligands are similar to those with true ligands. These figures reveal shared interaction types like **hydrogen bonding**, **$\pi-\pi$ stacking**, and **salt bridge**, depicted with color-coded dashed lines for each interaction type. This should help clarify the rationale and evidence supporting our approach.
>
> [1] Polizzi, Nicholas F., and William F. DeGrado. "A defined structural unit enables de novo design of small-molecule–binding proteins." Science 369, no. 6508 (2020)
>
> ## Response to ablation studies evaluating the impact of critical design choices
>
> Thanks for your advice on adding more ablation studies on design choices. We did ablation studies on the effectiveness of distributional alignment, where the major difference is that fragment sizes are modified. We found that aligned fragment sizes with real ligands could provide the best performance. The result is shown in **Table 5**.
>
> As for the distance thresholds, we define the pocket following the UniMol setup. We also found our model can also adapt to an 8Å setup in the ToughM1 experiment. Following your advice, we did an ablation study on the distance thresholds. We tested our method with three different thresholds: 4Å, 6Å, and 8Å. The result is shown below:
>
> |  | Kahraman$\uparrow$ | Tough M1$\uparrow$ | Fpocket$\downarrow$ | Druggability$\downarrow$ | Total SASA$\downarrow$ | Hydrophobicity$\downarrow$|
> | --- | --- | --- | --- | --- | --- | --- |
> | 4Å | 0.7062 | 0.7549 | 0.1240 | 0.1095 | 28.29 | 13.07 |
> | 6Å | 0.7870 | 0.8178 | 0.1238 | 0.1090 | 31.17 | 12.01 |
> | 8Å | 0.8322 | 0.8292 | 0.1256 | 0.1125 | 34.83 | 12.92 |
>
> Since the 8Å threshold corresponds with the pocket definition in the Kahraman and Tough M1 datasets, it leads to optimal results. Our decision to use a 6Å threshold was made to align with the methodology of pretraining data creation by Uni-Mol, facilitating a fair comparison. Notably, even with the 6Å threshold, we achieved strong results in the pocket-matching task, which serves as a testament to the effectiveness of our approach.
>
> ## Response to substantial discrepancies between pseudo and real ligands
>
> Thank you for pointing out the discrepancies between pseudo and real ligands. We understand and acknowledge your concern. To mitigate the impact of this discrepancy, we have made several efforts. First, we performed **distribution alignment** to make the data distribution of pseudo-ligands more similar to that of true ligands in PDBbind. Additionally, we **fixed the molecule encoder during pretraining** to prevent it from being misled by the discrepancy. Our ablation studies have shown that these strategies are effective:
>
> |  | Kahraman$\uparrow$ | Tough M1$\uparrow$ | Fpocket$\downarrow$ | Druggability$\downarrow$ | Total SASA$\downarrow$ | Hydrophobicity$\downarrow$|
> | --- | --- | --- | --- | --- | --- | --- |
> | ProFSA | 0.7870 | 0.8178 | 0.1238 | 0.1090 | 31.17 | 12.01 |
> | w/o alignment | 0.7614 | 0.7589 | 0.1265 | 0.1108 | 34.79 | 14.86 |
> | w/o fix mol encoder | 0.6905 | 0.7337 | 0.1247 | 0.1094 | 32.17 | 12.20 |
>
> The table above shows that lacking distribution alignment and not fixing the molecular encoder during pretraining both lead to lower performance, underscoring the effectiveness of our methods in reducing discrepancies.
>
> Theorem 3.1 theoretically supports our approach's efficacy, even with discrepancies between pseudo and real ligands. Empirically, our method outperforms other pretraining techniques, confirming its effectiveness despite these discrepancies.

---

> > ### Author Response · Authors · 2023-11-17
> >
> > # Response 2/2
> >
> > ## Response to the question on N Terminal and C Terminal
> >
> > When a peptide bond is formed (green), one amine group and one carboxy group are left untouched. Therefore, in a linear polypeptide, there would always be a free amine group and a free carboxy group, and they are called the N terminal(blue) and the C terminal(red) respectively in biochemistry. You can find the visualization in **Appendix H**.
> >
> > ## Response to fixing the molecule encoder in contrastive learning
> >
> > From an **effectiveness** standpoint, as you mentioned before, there are unavoidable discrepancies between pseudo and true ligands. If we didn't fix our molecule encoder during pretraining, then the conditions of Theorem 3.1 might not be met. The ablation results presented in the previous response clearly indicate a performance decline when the molecular encoder is not fixed during contrastive learning, particularly in the context of pocket matching. This supports our assertion that the pocket encoder struggles to acquire chemical knowledge without the stability provided by a fixed molecular encoder.
> >
> > From an **efficiency** standpoint, a fixed molecule encoder also offers significant advantages. It reduces memory usage and accelerates training speed because all molecule embeddings can be pre-computed before training. This eliminates the need for feed-forward and back-propagation operations of the molecule encoder during training, leading to efficiency gains. Additionally, adapting to different molecule encoders becomes more straightforward when they are not co-trained during the pretraining process. This approach allows for more flexibility and ease in integrating various encoding methodologies **even if the molecule encoder is not accessible for training**. That is to say, our approach can even be used with undifferentiable molecular representations.
> >
> > ## Response to extending to other tasks like protein-protein interaction prediction
> >
> > Thank you for pointing out the potential application of our framework to protein-protein interaction (PPI) tasks. We have to point out that the interactions between proteins and proteins are far different from proteins and molecules. Specifically, the interaction areas are usually much larger, but shallower, and involved functional groups are less diverse due to limited types of natural amino acids. Therefore, several changes should be made to extend our ProFSA method to PPI tasks:
> >
> > 1. As for data creation, unlike the localized and specific nature of pocket-ligand interactions, protein-protein interactions are **more global and involve broader protein features**. This requires a shift in the data creation process to split entire protein structures into domains instead of short fragments and to capture potential interactions between them.
> >
> > 2. The interaction interface between protein and protein is much larger. We might need to change from atomic-level to residue-level representations to efficiently capture interaction information.
> >
> > 3. For contrastive learning training, ProFSA uses a fixed pretrained molecular encoder to cope with various small molecules and their atomic representation. However, for protein-protein interaction, it is not necessary as we could only model 20 types of amino acids at the residue level.
> >
> > Despite several changes that need to be done for PPI tasks, we find that we could divide protein complexes into local interactions. When the complex structures and monomer structures are provided, we have achieved a reasonable result on a flexible PPI affinity prediction benchmark with our current model in a zero-shot way.
> >
> > | Method            | Spearman $\uparrow$  |
> > |-------------------|----------------------|
> > | SchNet            | 0.072 ± 0.021        |
> > | DimeNet++         | 0.171 ± 0.054        |
> > | EGNN              | 0.080 ± 0.038        |
> > | TorchMD           | 0.117 ± 0.008        |
> > | GET               | **0.363 ± 0.017**    |
> > | ProFSA(zero-shot) | _0.248_              |
> >
> > Surprisingly, as a **zero-shot** method, ProFSA is able to outperform other **supervised learning** models except for GET, a newly proposed unified model. This demonstrates our approach is also able to capture protein-protein interaction information despite it being designed for protein-ligand interaction.
> >
> > A detailed explanation of this experiment is shown in **Appendix C6**.

---

> ### Author Response · Authors · 2023-11-21
> **Explanation of the newly revised paper**
>
> In our revised paper, we have incorporated changes following feedback from another reviewer, which we believe also addresses your concerns and queries.
>
> Regarding your question on the evidence supporting our approach, particularly the use of peptides to simulate small molecules, we've enriched **Section 3.1**. A new paragraph there now provides an in-depth explanation of non-covalent interactions. To aid understanding, **Figure 2** has been updated with visualizations that include both real and pseudo receptor-ligand pairs, illustrating three types of interactions. Furthermore, we delve into the details of **how intra-protein interactions mirror protein-ligand interactions**, involving specific types of amino acids.
>
> To better evaluate our design choices, the ablation studies have been extended. Alongside the existing studies in **Table 4** (different pretrained molecular encoders), **Table 5** (distribution alignment), and **Figure 5** (data scale). we've added a study on the impact of using a fixed molecular encoder in **Table 5** in the main text to address your question on the fixed molecular encoder. We've also included an ablation study on varying **cutoff values for pocket definition**, placed in the **Appendix C.4** due to its more focused scope. We can move it to the main part of the paper in the future if needed.
>
> We acknowledge your concern about the **differences between true and pseudo ligands**, so we've expanded our discussion on the distinct properties of our curated dataset versus the PDBBind dataset in **Figure 9** and **Section 3.2**. The ablation study in **Table 5** demonstrates our methods and their effectiveness in minimizing these discrepancies. Notably, the performance significantly drops without our distribution alignment or fixed molecular encoder, highlighting our efforts to mitigate these issues. The updated **Figure 2** further supports this, showing that while the properties of true and pseudo ligands may differ, **the interaction types are consistent.** We hope these results can ease your concerns about our synthetic dataset, and they also respond to your query about **why we kept the molecular encoder constant during pocket pretraining.**
>
> We appreciate the constructive critiques and thoughtful engagement from you. Your feedback has undeniably contributed to the refinement and strengthening of our paper. We sincerely value the time and expertise invested in the review process, and we look forward to any further suggestions you may have.

---

### Official Review · Reviewer_kXw3 · 2023-10-31

**Soundness:** 3 good
**Presentation:** 3 good
**Contribution:** 3 good
**Rating:** 6
**Confidence:** 3

**Summary:**

This paper enhances protein pocket pretraining by introducing a new large pseudo ligand-pocket dataset. The dataset is constructed by segmenting a fragment from a protein and treating the neighboring area of the fragment as a pocket. Several important strategies are adopted to make the generated fragment-pocket pairs more like real ligand-pocket pairs. This results a dataset with 5.5 million pseudo ligand-pocket pairs. Contrastive learning is conducted using the generated dataset, in which a pretrained small molecular encoder is used to extract features for the fragments to align with a pocket encoder to be pretrained. Experiments are conducted on both pocket-only tasks and a pocket-molecule task.

**Strengths:**

1.The strategy of constructing pseudo ligand-pocket dataset is novel and has the potential to be extended to construct larger datasets.

2.Effective strategies are introduced to make the pesudo ligand-pocket pairs effective to mimic real ones and a practical contrastive learning strategy is adopted to address the difference between the segmented fragments from real ligands.

**Weaknesses:**

1. One weakness is that the proposed method is only evaluated on limited tasks.

2. The baselines in the  experiment are quite old, with the latest method published in 2020 except Uni-Mol.

**Questions:**

1. Will the proposed method work on other tasks, such as protein-ligand binding pose prediction?
2. Is there any new methods on the POCKET MATCHING task? If so, please include them in comparison.

---

> ### Author Response · Authors · 2023-11-17
>
> ## Response to limited tasks and potential in protein-ligand binding pose prediction
>
> In our paper, the primary objective is to assess the effectiveness of our pretraining method and the quality of the trained pocket representations. To achieve this, we focused on two types of downstream tasks: pocket-only tasks (pocket property prediction and pocket matching) and pocket-ligand interaction tasks (ligand binding affinity prediction).
>
> We appreciate the suggestion to apply our method to the protein-ligand binding pose prediction task. However, it's important to note that our current evaluation framework is designed to specifically assess **pocket representations**. For the tasks we chose, the architecture is straightforward: either a **zero-shot** evaluation or a **simple MLP** for mapping embeddings to predictions. In contrast, protein-ligand binding pose prediction often involves **complex methodologies**. For instance, state-of-the-art methods like DiffDock[1] require training a diffusion generative model and a separate confidence model, while other approaches like EDM-Dock[2], rely on reconstructing ligand conformations from predicted distance maps. These methods are not end-to-end and do not directly align with our objective of evaluating pretrained pocket representations.
>
> Nevertheless, we recognize the potential of our method in enhancing existing binding pose prediction techniques. To integrate our approach, we would need to modify our framework. Using the same data creation strategy, we could train a binding pose prediction model with our preprocessed data, which could then be further fine-tuned using real pocket-ligand pair data from sources like PDBbind. Additionally, our data could be used to train a side-chain packing model, allowing for side-chain flexibility during docking. Thank you again for your advice, and we will leave protein-ligand binding pose prediction as a separate future work for our method.
>
> In response to the need for evaluating our method on a broader range of tasks, we have extended our analysis to include two additional downstream tasks: LEP (Ligand Efficacy Prediction), and PPA (Protein-Protein Affinity Prediction).
>
> result for LEP:
>
> | Method      | AUROC $\uparrow$ | AUPRC $\uparrow$ |
> |-------------|--------------|--------------|
> | ATOM3D-GNN  | 0.681        | 0.598        |
> | GeoSSL      | 0.776±0.03   | 0.694±0.06   |
> | Uni-Mol     | 0.782±0.02   | 0.695±0.07   |
> | ProFSA      | 0.840±0.04   | 0.806±0.04   |
>
> GeoSSL and Uni-Mol, both pretraining methods, yield comparable results. However, ProFSA outperforms these methods, demonstrating the advantage of our pocket pretraining approach.
>
> result for PPA:
>
> | Method            | Spearman $\uparrow$  |
> |-------------------|----------------------|
> | SchNet            | 0.072 ± 0.021        |
> | DimeNet++         | 0.171 ± 0.054        |
> | EGNN              | 0.080 ± 0.038        |
> | TorchMD           | 0.117 ± 0.008        |
> | GET               | **0.363 ± 0.017**    |
> | ProFSA(zero shot) | _0.248_              |
>
> As a **zero-shot** method, ProFSA is able to outperform other **supervised learning** models except for GET, a newly proposed unified model. This demonstrates our approach is able to capture protein-protein interaction information despite it being designed for protein-ligand interaction modeling.
>
> You can find detailed experiment settings and results in **Appendix C5 and C6**.
>
> [1] Corso et al., "Diffdock: Diffusion steps, twists, and turns for molecular docking.", ICLR 2023.
> [2] Masters et al., "Deep learning model for efficient protein–ligand docking with implicit side-chain flexibility." Journal of Chemical Information and Modeling 63, no. 6 (2023).
>
> ## Response to the issue of lack of latest baselines
>
> For the pocket matching task, CoSP is a newly proposed baseline which is published in ECML PKDD 2023. Alongside CoSP and Uni-Mol, we selected some of the most effective baseline results presented in the CoSP paper. We also tried to evaluate recent methods like PocketAnchor (Li et al., Cell Systems 2022) on our own since they are not tested on the pocket matching task. However, since the result didn't outperform other baseline machine learning methods(Uni-Mol, CoSP, and DeeplyTough), we decided not to include it in our final baseline comparison. We can include it in the camera-ready version if needed.
>
> For the ligand binding affinity task, a lot of newly proposed baselines are included, i.e. ProNet (Wang et al., 2022b); as well as pretraining methods such as GeoSSL (Liu et al., 2023), EGNN-PLM (Wu et al., 2022), DeepAffinity (Karimi et al., 2019) and Uni-Mol (Zhou et al., 2023).

---

> ### Author Response · Authors · 2023-11-22
> **Anticipating your response**
>
> Thank you for your insightful feedback. In response to your concerns regarding the scope of our tasks, we have expanded our research in the revised paper. We now include additional tasks such as **ligand efficiency prediction** and **protein-protein affinity prediction**, detailed in **Appendix C5 and C6**. These tasks were incorporated to enhance the robustness of our method. We acknowledge that they are currently in the appendix due to space constraints, but we are open to relocating them to the main text if it would be beneficial.
>
> Regarding your concern about the lack of recent baselines, we provided a detailed explanation in our previous response. We want to reiterate that our benchmarks do include the latest baselines, with some as recent as 2023.
>
> We greatly appreciate the time and effort you have invested in reviewing our work. Your feedback has been invaluable in refining our research. Please let us know if there are any other aspects of our paper that you would like us to address or clarify. We look forward to your further feedback.

---

### Official Review · Reviewer_FNxz · 2023-11-01

**Soundness:** 3 good
**Presentation:** 3 good
**Contribution:** 3 good
**Rating:** 6
**Confidence:** 1

**Summary:**

This paper proposes a novel approach called ProFSA for pretraining pocket representations based on the guided fragment-surroundings contrastive learning. Furthermore, a novel scalable pairwise data synthesis pipeline is designed to extract pseudo-ligand-pocket pairs from protein-only data. Extensive experiments demonstrate the potential of ProFSA as a powerful tool in the field of drug discovery.

**Strengths:**

1. The paper is well-structured and clearly articulates the research methodology and findings. The overall presentation is easy for readers to grasp the key ideas of this paper.

2. By utilizing pseudo-ligand construction and pocket construction, authors develop an innovative strategy for mining extensive protein-only data from the PDB repository, which can effectively alleviate the scarcity of experimentally determined pocket-ligand pairs.

3. A contrastive learning approach in the protein-fragment space is introduced to attain ligand-aware pocket representations. By sampling negative samples from protein pockets and pseudo-ligands, the pocket encoder can learn to identify the true positive sample when given the other one.

4. Extensive experiments demonstrate the potential of ProFSA as a powerful tool in the drug discovery field.

**Weaknesses:**

1. I'm not fully satisfied with the Related Work section. More work should be presented, such as [1], [2] and [3].

2. Why is COSP introduced as a component of the pocket pretraining method in Section 2.2, but not included as a baseline in Table 3?

3. In section 3.2, the authors mention that the "the first loss is to differentiate the corresponding ligand fragment from a pool of candidates for a given pocket." The first loss is constructed by sampling negative samples from protein pocket. Therefore, I think the purpose of the first loss is to identify the true protein pocket when given a pseudo-ligand.

4. I am confused about how ProFSA works without the distributional alignment  mechanism. In this context, what determines the length of the pocket representation?

[1] Liu S, Guo H, Tang J. Molecular geometry pretraining with se (3)-invariant denoising distance matching[J]. arXiv preprint arXiv:2206.13602, 2022.
[2] Wu F, Li S, Wu L, et al. Discovering the representation bottleneck of graph neural networks from multi-order interactions[J]. arXiv preprint arXiv:2205.07266, 2022.
[3] Karimi M, Wu D, Wang Z, et al. DeepAffinity: interpretable deep learning of compound–protein affinity through unified recurrent and convolutional neural networks[J]. Bioinformatics, 2019, 35(18): 3329-3338.

**Questions:**

Please see the questions in weaknesses.

---

> ### Author Response · Authors · 2023-11-17
> **Question about the confidence score**
>
> # Question about the confidence score
>
> Thanks for your review. We found your understanding and judgment of our paper comprehensive and precise. Also, you provided detailed and thoughtful advice for our paper, which helped a lot. We sincerely appreciate your review, but we also wonder if there is any misunderstanding here, as you only give a confidence score of 1.

---

> ### Author Response · Authors · 2023-11-17
>
> ## Response to lack of related works
>
> Thanks for providing us with more insightful related works that could support our arguments. We actually have cited these three papers in the section on ligand binding affinity experiments, but we are also happy to add these in Related Works section 2.2. You can find the change in pdfdiff.
>
> ## Response to lack of COSP as a baseline in Table 3
>
> Because best to our knowledge, they did not release their code and they did not test their method on the ligand binding affinity dataset.
>
> ## Response to the question on the loss terms
>
> Thank you for your advice. We apologize that we accidentally mentioned loss1 and loss2 in reverse, and we are sorry for the confusing statement. You are correct that the first loss is to identify the true protein pocket when given a pseudo-ligand. We revise the original statement to: "The primary purpose of the first loss is to identify the true protein pocket from a batch of samples when given a pseudo-ligand. Similarly, the second loss seeks to identify the corresponding ligand fragment for a given pocket." You can also find the change in pdfdiff.
>
> ## Response to the question on the length of the pocket representation without alignment
>
> With or without distributional alignment, pockets are always defined by the given protein fragment with a fixed distance cutoff (6Å in our works, following the UniMol setup). The distributional alignment process merely samples these pairs to match the sizes of real ligands and pockets. Without distributional alignment, fragments are uniformly sampled from 1 to 7 residues. Pockets are similarly determined as with residues within the range of 6Å around peptide fragments.

---

> ### Author Response · Authors · 2023-11-22
> **Anticipating your response**
>
> Thank you for your invaluable feedback on our paper. In response to your insightful suggestions, we have made several modifications in our revised paper.
>
> We have cited additional papers in Section 2.2 following your advice. The description of loss terms in Section 3.2 has been corrected.  COSP was not included as a baseline for the Ligand Binding Affinity (LBA)  task because it is not open-sourced, and its performance on LBA was not evaluated in their published work. We used it as a baseline in the pocket-matching task.
>
> In Section 3, we've expanded our discussion to answer your question on **ProFSA's effectiveness without distributional alignment**. This section now offers a detailed explanation of the foundational concepts and justifications for our method. We draw attention to the parallels between ligand-protein and intra-protein non-covalent interactions, as shown in **Figure 2**. This comparison supports our strategy of using peptides as stand-ins for actual ligands to mimic pocket-ligand interactions. Consequently, **even in the absence of distributional alignment, the types of interactions remain comparably relevant**, which explains why ProFSA continues to function, albeit with reduced effectiveness, as demonstrated in Table 5. We also want to clarify that distributional alignment is only applied to fragment-pocket complex as a whole, and **it would not change the definition of the pocket with a given ligand**. Without this alignment, these fragments are chosen uniformly, with lengths varying from 1 to 7 residues, and pockets are consistently identified based on residues within a 6Å radius of these peptide fragments.
>
> We are grateful for the time and effort you have dedicated to reviewing our work. Your thorough and constructive feedback has significantly contributed to the refinement of our research. Please let us know if there are any other aspects of our paper that you would like us to address or clarify. We look forward to your further feedback.

---

### Official Review · Reviewer_yfM3 · 2023-11-01

**Soundness:** 3 good
**Presentation:** 4 excellent
**Contribution:** 3 good
**Rating:** 6
**Confidence:** 5

**Summary:**

This paper primarily aims to enhance the pocket pretraining method, as existing approaches only consider pockets during pretraining. There are two main contributions in this paper: (1) The authors introduce a novel method, ProFSA, for pocket pretraining, which extracts additional information from corresponding ligands. However, the number of pocket-ligand complex structures is quite limited in existing datasets. (2) To address this issue, the authors generate over 5 million complexes by segmenting fragments and their corresponding pockets in protein structures. By aligning features of fragments and pockets, the pocket encoder learns the interaction between fragments and pockets. The authors design downstream tasks such as pocket druggability prediction, pocket matching, and ligand binding affinity prediction to demonstrate the effectiveness of ProFSA.

**Strengths:**

The authors propose a new perspective of pretraining pockets and construct a large-scale dataset, which data distribution is also considered, to make the efficient pre-training possible.

The results are competitive, especially for zero-shot settings.

Abundant experiments and ablation study support the argument and result of the authors.

**Weaknesses:**

1. The technical novelty is limited.
  - The pocket encoder is borrowed from Uni-Mol.
  - The contrastive loss is the vanilla form of classical contrastive learning.

2. The bound of Theorem 3.1 is trivial. The authors claim that the bound naturally exists for these representations extracted by pretrained molecule models. However, it's a bit counterintuitive, because many models not pretrained on molecule datasets also fulfill this prior. So, can these models be used for this task? **I strongly suggest removing this part from the paper**.

3. Some issues about dataset creation:
 - 3.1. The authors consider the distribution of ligand size and pocket size when designing the dataset. However, molecules possess more properties that can also lead to imbalance. It would be better to, at least, add some discussion about this issue.
 - 3.2. In the second stage of the data construction process, the approach to defining pockets needs further explanation or an ablation study.

4. Experiments: It would be better to add some biological justification or visualization of the results.

For this paper, one fact is that the technical novelty is below the bar of ICLR. However, I admire the simple but effective model for the right question. It's a struggle for me to make a decision. I will maintain a neutral attitude and make my final decision after the discussion.

**Questions:**

See weakness.

---

> ### Author Response · Authors · 2023-11-17
>
> # Response 1/3
> We appreciate the time and effort you have dedicated to reviewing our paper, and we are grateful for your constructive feedback and thoughtful evaluation of our work. We would like to address your comments about the technical novelty of our work and provide additional clarification on certain aspects of our paper.
>
> ## Response to technical novelty
>
> Firstly, we acknowledge your observation that we didn't use any fancy models compared with typical **model-centric** works. While we respect your assessment, we would like to highlight that our primary focus in this paper is a **data-centric** pretraining method to introduce groundbreaking improvements in the field of protein pocket pretraining. Our work aims to address a critical challenge of data scarcity in the field of protein pocket representation by constructing large-scale synthetic data that facilitates the pretraining of models, ultimately enhancing the accuracy and robustness of protein pocket representations. We believe that the research community can conduct fast following-ups on our **released dataset** with more sophisticated models, and even extend our pipelines to other tasks like docking or drug design. We also would like to point out that we have provided a novel method to **distill knowledge from well-trained molecule models** to protein models. Though employing the same contrastive loss, our approach is different from existing contrastive learning models since it uses a fixed molecular encoder. The motivation here is to use a well-trained molecular encoder on a relatively larger dataset to **guide** the training of the protein encoder. For example, some quantum-chemistry properties that are difficult to compute for large systems like proteins could be distilled from molecules, by using some quantum-chemistry-aware molecular encoders such as Frad. Though our framework seems to be simple, it is non-trivial to make it work. We have made several efforts to solve the unavoidable discrepancies between true ligands and pseudo ligands. The ablation study shows that our efforts are effective and necessary:
>
> |  | Kahraman$\uparrow$ | Tough M1$\uparrow$ | Fpocket$\downarrow$ | Druggability$\downarrow$ | Total SASA$\downarrow$ | Hydrophobicity$\downarrow$|
> | --- | --- | --- | --- | --- | --- | --- |
> | ProFSA | 0.7870 | 0.8178 | 0.1238 | 0.1090 | 31.17 | 12.01 |
> | w/o alignment | 0.7614 | 0.7589 | 0.1265 | 0.1108 | 34.79 | 14.86 |
> | w/o fix mol encoder | 0.6905 | 0.7337 | 0.1247 | 0.1094 | 32.17 | 12.20 |
>
> Regarding your positive acknowledgment of our "simple but effective model for the right question," we are pleased to hear that our approach resonates with the objective we set out to achieve. We designed our model with **simplicity** in mind, prioritizing effectiveness and practical utility for the specific problem domain. This deliberate choice aligns with the notion that sometimes the most impactful solutions are elegantly straightforward. Also, we intentionally borrowed a pocket encoder from Uni-Mol to make a **fair comparison** with it, which strongly supports the power of our dataset. Similar to the molecule encoder, our pocket encoder can be changed to any other model due to our flexible framework. Notably, **we didn't load the pretrained weights of Uni-Mol pocket encoder**. We only use the same backbone architecture and the pretraining was completely done on our processed data with our training strategy.
>
> Notably, many simple but effective approaches have been recognized by top-tier conferences and journals. A prime example of this is in the field of protein language models, like the ESM series and ProtTrans. These studies adapted the Transformer architecture and masked language modeling techniques from natural language processing to protein sequences. While they didn't introduce groundbreaking techniques, their substantial contributions to protein modeling are evident, with publications in prestigious conferences and journals like NeurIPS, ICML, TPAMI, PNAS, and Science. Another example is the widely acclaimed CLIP paper presented at ICML 2021. CLIP, while not employing novel techniques, stands as a hallmark of data-centric deep learning. Its use of contrastive learning enabled training on expansive web-sourced text-image datasets, moving beyond the constraints of meticulously curated databases like MS-COCO. This aligns with our approach to addressing **data scarcity challenges** by facilitating training on large-scale datasets, mirroring our strategy for overcoming similar hurdles in the protein pocket modeling domain.
>
> We appreciate your efforts to remain neutral and understand the challenges in decision-making. We believe further discussion will highlight our paper's contributions. Your feedback and insights in the upcoming discussion will be invaluable for refining our work.

---

> > ### Author Response · Authors · 2023-11-17
> >
> > # Response 2/3
> >
> > ## Response to the bound of Theorem 3.1
> >
> > We are not very clear about the point of your question. We have two understandings and they are replied as follows.
> >
> > If you mean $ ||t − t ^{(0)}|| < 1/2 l_T$ is trivial, please note that it is only a condition of the theorem. As you said, many other encoders can also satisfy this condition. This shows that our method is applicable to many different molecular encoders with or without pretraining as we proved in **Table 4**. Moreover, the condition is only a necessary condition. We also need the pretraining contrastive loss to be sufficiently small to guarantee a small contrastive loss between pockets and real ligands. In fact, in the process of our early exploration, we noticed that some encoders cannot achieve a low pretraining loss, indicating the entire conditions of the theorem are nontrivial.
> >
> > If you mean that the conclusion of theorem 3.1 is trivial, we want to emphasize that our conclusion is $\lim_{L_i(t, s) =0}L_i(t^{(0)}, s) =0$. (We write it in a $\epsilon$-$\delta$ language in paper. They are equivalent forms.) Our conclusion shows the loss containing real ligands is consistent with the loss containing pseudo ligands that we optimized in pre-training.
> >
> > In contrast,  a trivial result from $||t − t ^{(0)}|| < 1/ 2l_T$ is that $\lim_{L_i(t, s) =0}L_i(t^{(0)}, s) <B$, B is a bound related to l_t and the representations. It does not guarantee a consistent loss and is substantially different from our result.
> > Therefore, our theorem is nontrivial and reveals the transfer ability of our contrastive pre-training from the pseudo-ligand domain to the real ligand domain.
> >
> > As for this theoretical result, we are open to further discussion.
> >
> > ## Response to molecules possess more properties that can also lead to imbalance
> >
> > It is correct that molecules have more properties than their size, like logP, Hbond donor and acceptor number, and rotatory bond number. As small molecule drugs are mostly designed to penetrate barriers like gut or cell membranes, they usually have much larger logP values, which means more hydrophobic. For the same reason, they usually have fewer hydrogen bond donors and acceptors, and only minimal essential ones are kept for specified interactions. Also, to minimize the entropy effect upon binding and to increase binding affinity, rotatory bonds are also unfavored. However, it is impossible to mimic such features with peptides, as the backbone of peptides is intrinsically hydrophilic and flexible. As we showed in a new figure, our pseudo-ligands are less similar to real ligands in those properties even with size alignment. You can find the figure in the new pdf in **Appendix G**.
> >
> > We believe that the aforementioned divergence is the main limitation of poor zero-shot performance in predicting hydrophobicity scores (**Table 1**). However, as demonstrated in previous publications like CoSP, we could leverage real ligand-pocket pairs from PDBBind or BioLip database to further finetune our network, as an extension of the pipeline. In this way, we could handle property mismatches but still enjoy the power of our large-scale pretraining.
> >
> > ## Response to the approach to defining pockets
> >
> > We define the pocket following the UniMol setup. We found our model can also adapt to an 8Å setup in the toughM1 experiment
> >
> > In particular, we define the pocket for each protein-ligand pair as residues of the protein that have at least one atom within the range of 6Å from a heavy atom in the ligand. To further explain our design, we have done an ablation study on different choices of thresholds of 4Å, 6Å, and 8Å:
> >
> > |  | Kahraman$\uparrow$ | Tough M1$\uparrow$ | Fpocket$\downarrow$ | Druggability$\downarrow$ | Total SASA$\downarrow$ | Hydrophobicity$\downarrow$|
> > | --- | --- | --- | --- | --- | --- | --- |
> > | 4Å | 0.7062 | 0.7549 | 0.1240 | 0.1095 | 28.29 | 13.07 |
> > | 6Å | 0.7870 | 0.8178 | 0.1238 | 0.1090 | 31.17 | 12.01 |
> > | 8Å | 0.8322 | 0.8292 | 0.1256 | 0.1125 | 34.83 | 12.92 |
> >
> > Since the 8Å threshold corresponds with the pocket definition in the Kahraman and Tough M1 datasets, it leads to optimal results. Our decision to use a 6Å threshold was made to align with the methodology of pretraining data creation by Uni-Mol, facilitating a fair comparison. Notably, even with the 6Å threshold, we achieved strong results in the pocket-matching task, which serves as a testament to the effectiveness of our approach.

---

> > > ### Author Response · Authors · 2023-11-17
> > >
> > > # Response 3/3
> > >
> > > ## Response to biological justification and visualization of the results
> > >
> > > Our method's **biological justification** for achieving good experimental results is based on the properties and interactions of pseudo-ligands. As discussed in our paper, these pseudo-ligands are similar in size to real ligands and engage in comparable non-covalent interactions with protein pockets. This fundamental similarity is crucial for the effectiveness of our approach.
> > >
> > > Moreover, our methodology is further supported by the findings in the 2020 Science paper titled "A defined structural unit enables de novo design of small-molecule–binding proteins."[1] This research underscores that **intra-protein interactions are analogous to protein-ligand interactions**, which validates our use of peptides as proxies for small molecules. By employing peptides as pseudo-ligands in this manner, our pocket encoder is able to learn and replicate the interaction dynamics typically observed in real ligand scenarios. This understanding is pivotal to the success of our method in downstream applications.
> > >
> > > To better justify our methods and results, following your advice, we also provide some visualizations in **Appendix F**. **Figure 6** illustrates that the pseudo pairs we created share various interaction types commonly found in real pocket-ligand pairs, such as **hydrogen bonding, $\pi-\pi$ stacking, and salt bridge**. In the figures, each type of interaction is represented by dashed lines, color-coded to correspond with the specific interaction type.
> > >
> > > Another visualization of our pocket-matching result is shown in **Figure 7**. We showed an example of two non-homology ATP binding pockets to explain why pocket-matching can benefit from interaction-aware pretraining so that we have achieved superior results in the Kahraman dataset and the BioLip t-SNE visualization. The PEP carboxykinase(PDB:1AYL) and tRNA synthetase(PDB:1B8A) are two ATP-binding proteins that share zero sequence similarity (verified with the BLAST) extracted from the Kahraman dataset. However, as they are both fueled by ATP, their binding site shares similar binding patterns. The cation-$\pi$ interactions(blue dash lines) and salt bridges (magenta dash lines) are important to the ATP binding, which can be viewed as convergent evolutions at the molecule level. Even though, these two pockets are very distinct in terms of shapes and sizes because they bind ATPs in different conformations. Therefore, biochemical interactions are the key to accomplishing the pocket-matching task, which is ignored in previous self-supervised learning methods like Uni-Mol.
> > >
> > > [1] Polizzi, Nicholas F., and William F. DeGrado. "A defined structural unit enables de novo design of small-molecule–binding proteins." Science 369, no. 6508 (2020)

---

> > > > ### Comment · Reviewer_yfM3 · 2023-11-18
> > > > **Biological justification**
> > > >
> > > > For the visualization, I mean you need include some examples in the main part, not just in the appendix. If there are some cases where your model performs well while the other models do not, that would be more convincing.

---

> > > ### Comment · Reviewer_yfM3 · 2023-11-18
> > > **Thank you for your response.**
> > >
> > > Sorry for the confusion. Regarding the theorem part, I mean it is not necessary to involve Thm. 3.1 to decorate your models, simply removing it should be fine.

---

> ### Comment · Reviewer_yfM3 · 2023-11-18
> **About the revisions.**
>
> By now, I haven't found your revised version of your paper (Correct me if I'm wrong).
> If in the revision, you can focus more on the problem & data rather than the model & theorem and provide more visualization cases that can help the machine learning community better understand the pocket-ligand prediction problem. I would raise my score.
>
> Additionally, since your model involves the pretrained mol encoder,  the existing pretraining approaches on molecules should be extensively discussed in the related works.

---

> > ### Author Response · Authors · 2023-11-20
> >
> > We sincerely appreciate the thoughtful feedback provided on our paper and have diligently incorporated your suggestions into this revised version. In response to your guidance, we have made three significant modifications to the core content of our paper, all of which aim to enhance clarity and address the concerns raised.
> >
> > Taking into account your suggestion, we have removed **Theorem 3.1** from the main body of our paper. Additionally, we've improved the explanation and included an ablation study on our method of defining pockets. This ablation study, focusing on the cutoff values and primarily a hyperparameter issue, is placed in the Appendix due to its more limited scope of insight. We can move it to the main body of the paper in the future if needed. We have also expanded our discussion on the differences between our dataset and the PDBBind dataset, elucidating how fixed molecule encoders contribute to mitigating this discrepancy.
> >
> > In **Section 3.1**, we have introduced a new paragraph dedicated to providing readers with a deeper understanding of non-covalent interactions. To facilitate comprehension, we have included visualizations featuring three types of interactions, incorporating both real and pseudo receptor-ligand pairs in the new **Figure 2**. Furthermore, we delve into the details of how intra-protein interactions mirror protein-ligand interactions, involving specific types of amino acids.
> >
> > The biological justifications for the efficacy of interaction-aware pretraining are now more extensively explored in our revised paper. Specifically, we present a compelling showcase featuring estradiol-binding proteins, offering insights into the geometric disparities between the compared proteins and highlighting the similarities in their binding interfaces in the new **Figure 4a**. The existing visualization of BioLip pocket representations in the new **Figure 4b** can make our claims more convincing. Together, these visual aids illustrate how our interaction-aware pretraining empowers models to focus on crucial interface residues, discern subtle distinctions, and disregard irrelevant geometric dissimilarities.
> >
> > We have also included an in-depth discussion of pretrained molecule encoders in the newly added **Section 2.3**, further enriching the theoretical foundation of our work.
> > All changes can be found in the renewed paper, which is updated lively.
> >
> > We genuinely value the constructive input you provided during the review process, as it has undoubtedly contributed to the refinement of our paper. Your insights have proven invaluable in elevating the clarity and depth of our contributions to the field. We express our gratitude for the time and expertise invested in reviewing our work, and we look forward to any additional feedback or suggestions you may have.

---

### Author Response · Authors · 2023-11-23

Dear AC and reviewers,

We sincerely appreciate your time and efforts in reviewing our work. Based on your suggestions, we have revised our paper. We would like to use this section to reiterate the explanation for some common concerns and summarize the contributions of our paper.

In our revised paper, we provide a detailed explanation of the motivation and theory behind our method: **the mechanisms of pocket-peptide interactions and pocket-ligand interactions are very similar**. Our proposed **data-centric** pretraining pipeline is unique in its use of peptides to mimic small molecules, a method that is, to our knowledge, **the first of its kind**. We expanded our discussion about its biological insights in **Section 3**. We add a **new Figure 2 to illustrate different types of non-covalent interactions** that are common for both ligand-protein data and intra-protein data. Following this theory, we have designed a pipeline to generate **over 5 million** synthetic ligand-protein pairs, which greatly eases the **data scarcity** problem in the field of protein-ligand interaction learning. We also prove our data quite effective in various downstream tasks, including pocket property prediction, pocket matching, ligand binding affinity prediction, ligand efficiency prediction, and protein-protein interaction tasks. The **last two tasks are newly added** to further support our method (**Appendix C5 and C6**). We are confident that our **ready-to-release synthetic dataset** will significantly benefit the AI for drug discovery community. Its impact extends beyond enhancing pocket representations in downstream analyses, as it also equips researchers with a valuable resource for a range of applications, such as protein-ligand docking, structure-based drug design, and virtual screening. Importantly, these areas often face challenges due to **a lack of training data necessary for learning effective binding patterns between protein pockets and ligands**.

In our revised paper, we recognize the differences between real ligands and pseudo ligands (peptides). We address these discrepancies and present a distribution plot of various properties in **Appendix G**. Our approach primarily leverages the **analogous nature of ligand-pocket and peptide-pocket interactions**. Therefore, despite these differences, our model can **still extract valuable interaction information from the pretraining dataset**, which enhances its performance in downstream tasks. This is illustrated in **Figure 4(a)**, showing the benefits of interaction-aware pretraining. We have also undertaken measures like distribution alignment and freezing the molecular encoder to mitigate the impact of these discrepancies. The positive results of these efforts, as evidenced in **Table 5**, confirm their effectiveness and necessity. Notably, our pretrained model can be **further finetuned with real labeled ligand-pocket pairwise data**, as we show in the Ligand Binding Affinity task in **section 4.3**.

As for other questions, we have added more ablation studies on our design choices, including the impact of a fixed molecular encoder (Table 5), and different threshold values to define pockets(Appendix C.4).

We are immensely grateful for the invaluable feedback from all the reviewers, which has guided us in refining and clarifying our work. We hope that the revised version of our paper, coupled with the discussion period, will more clearly highlight the novelty, effectiveness, and contributions of our approach to the field.

---

### Meta-Review · Area_Chair_dH6M · 2023-12-14

**Metareview:**

This paper introduces a novel avenue for training protein-ligand binding models, based on careful construction of a training dataset through breaking down peptides into fragments, which are then used as a pre-training task for a later protein-ligand binding model. The paper went through quite some revision in response to reviewers. There was some concern from reviewers that the ML models are fairly straightforward, but overall the experimental results are well-supported and the synthetic dataset will likely have independent value.

**Justification For Why Not Higher Score:**

Paper is good (clear consensus to accept), but no reviewers were advocating strongly.

**Justification For Why Not Lower Score:**

All reviewers argued to accept, and the paper has seen many of the potential issues addressed in rebuttal.

---

### Decision · Program_Chairs · 2024-01-16

Accept (poster)